

# A comparative study of the performance of ten metaheuristic algorithms for parameter estimation of solar photovoltaic models

Adel Zga[1], Farouq Zitouni[1], Saad Harous[2], Karam Sallam[2], Abdulaziz S. Almazyad[3], Guojiang Xiong [4] and Ali Wagdy Mohamed[5,6]

[1] Department of Computer Science and Information Technology, Laboratory of Artificial Intelligence and Information Technology, Kasdi Merbah University, Ouargla, Algeria
[2] Department of Computer Science, College of Computing and Informatics, University of Sharjah, Sharjah, United Arab Emirates
[3] Department of Computer Engineering, College of Computer and Information Sciences, King Saud University, Riyadh, Saudi Arabia
[4] Guizhou Key Laboratory of Intelligent Technology in Power System, College of Electrical Engineering, Guizhou University, Guiyang, China
[5] Operations Research Department, Faculty of Graduate Studies for Statistical Research, Cairo University, Giza, Egypt
[6] Applied Science Research Center, Applied Science Private University, Amman, Jordan

Corresponding author
Farouq Zitouni,
zitouni.farouq@univ-ouargla.dz

## ABSTRACT

This study conducts a comparative analysis of the performance of ten novel and well-performing metaheuristic algorithms for parameter estimation of solar photovoltaic models. This optimization problem involves accurately identifying parameters that reflect the complex and nonlinear behaviours of photovoltaic cells affected by changing environmental conditions and material inconsistencies. This estimation is challenging due to computational complexity and the risk of optimization errors, which can hinder reliable performance predictions. The algorithms evaluated include the Crayfish Optimization Algorithm, the Golf Optimization Algorithm, the Coati Optimization Algorithm, the Crested Porcupine Optimizer, the Growth Optimizer, the Artificial Protozoa Optimizer, the Secretary Bird Optimization Algorithm, the Mother Optimization Algorithm, the Election Optimizer Algorithm, and the Technical and Vocational Education and Training-Based Optimizer. These algorithms are applied to solve four well-established photovoltaic models: the single-diode model, the double-diode model, the triple-diode model, and different photovoltaic module models. The study focuses on key performance metrics such as execution time, number of function evaluations, and solution optimality. The results reveal significant differences in the efficiency and accuracy of the algorithms, with some algorithms demonstrating superior performance in specific models. The Friedman test was utilized to rank the performance of the various algorithms, revealing the Growth Optimizer as the top performer across all the considered models. This optimizer achieved a root mean square error of 9.8602187789E−04 for the single-diode model, 9.8248487610E−04 for both the double-diode and triple-diode models and 1.2307306856E−02 for the photovoltaic module model. This consistent success indicates that the Growth Optimizer is a strong contender for future enhancements aimed at further boosting its efficiency and effectiveness. Its current performance suggests significant potential for improvement, making it a promising focus for ongoing development efforts. The findings contribute to the understanding of the applicability and performance of metaheuristic algorithms in

renewable energy systems, providing valuable insights for optimizing photovoltaic models.

## INTRODUCTION

Optimization is the process of making systems or decisions as effective, efficient, or functional as possible. It is a fundamental concept that influences a broad range of everyday activities and operations. By applying optimization techniques, individuals and organizations can achieve the best possible outcomes with the least waste of time, effort, and resources. In daily life, optimization manifests in various forms, such as planning the quickest route to work to minimize commuting time, organizing schedules to enhance productivity, or managing household budgets to maximize financial resources.

Optimization techniques and algorithms play a pivotal role in various sectors, driving efficiency and enhancing the quality of solutions. In logistics and supply chain management (*Griffis, Bell & Closs, 2012*), optimization algorithms are employed to streamline routes and manage inventory levels effectively, thereby reducing costs and improving delivery times. In finance (*Katib et al., 2023*; *Boloş, Bradea & Delcea, 2021*), these techniques are crucial for portfolio optimization, risk management, and algorithmic trading, ensuring that investments yield the best possible returns with minimal risk. The manufacturing industry (*Para, Del Ser & Nebro, 2022*; *Fathi & Ghobakhloo, 2020*) leverages optimization to enhance production processes, minimize waste, and improve product quality. Similarly, in telecommunications (*Moscholios et al., 2022*; *Moradi et al., 2024*), optimization is used to design efficient networks, manage traffic flow, and ensure reliable connectivity.

Metaheuristics, such as genetic algorithms (*Holland, 1992*), particle swarm optimization (*Kennedy & Eberhart, 1995*), and simulated annealing (*Bertsimas & Tsitsiklis, 1993*), are widely favoured over exact methods for tackling complex optimization problems. These metaheuristic algorithms can be broadly categorized based on their sources of inspiration, which generally fall into four main groups: evolutionary algorithms, swarm intelligence, physics-based methods, and human-based methods (*Zitouni, Harous & Maamri, 2020*). Evolutionary algorithms, such as genetic algorithms and evolutionary mating algorithm (*Sulaiman et al., 2023*), are inspired by natural selection processes, while swarm intelligence methods, like particle swarm optimization and the archerfish hunting optimizer (*Zitouni et al., 2022*), mimic the collective behaviour in biological populations. Physics-based methods, such as simulated annealing and Young's double-slit experiment optimizer (*Abdel-Basset et al., 2023a*), draw from natural phenomena in physics. Finally, human-based methods simulate human decision-making strategies and include techniques like the human conception optimizer (*Acharya & Das, 2022*) and the

leader-advocate-believer-based optimization algorithm (*Reddy et al., 2023*) and other approaches inspired by human problem-solving behaviours. While exact methods such as linear programming (*Best & Ritter, 1985*) or branch and bound (*Lawler & Wood, 1966*) provide precise solutions, they often become computationally infeasible for large-scale or intricate problems. Metaheuristics, in contrast, offer a practical approach by delivering near-optimal solutions within a reasonable amount of time, making them suitable for real-world applications where time and resources are limited (*Blum & Roli, 2003*).

One particularly challenging optimization problem is the parameters' estimation of solar photovoltaic (PV) models (*Alsadi & Khatib, 2018*). As the world increasingly turns to renewable energy sources to reduce dependence on fossil fuels and combat climate change, optimizing PV models becomes essential. PV model optimization involves maximizing energy output and efficiency, which can significantly impact the viability and scalability of solar energy solutions. Effective optimization of PV systems ensures that solar panels operate at their highest potential, leading to greater energy savings and contributing to more sustainable future energy. This optimization not only supports environmental sustainability but it enhances energy security and reduces energy costs for consumers, thereby playing a crucial role in our daily lives (*Gu, Xiong & Fu, 2023*). Even so, a significant obstacle to the broader adoption of solar energy is its low conversion efficiency, which drives the need for more innovative methods to enhance the design of solar energy conversion systems. The solar cell serves as the essential element of a PV system, and accurately modelling and estimating its parameters is crucial for effective simulation, design, and control to achieve optimal performance. Estimating a solar cell's unknown parameters is complex due to the nonlinearity and multimodal nature of the search space. Traditional optimization techniques are often inefficient in this context because they struggle to navigate the complex, high-dimensional search space, making it easy to become trapped in local optima and preventing a comprehensive exploration of possible solutions. In contrast, metaheuristic algorithms are seen as promising due to their ability to explore the search space more effectively, avoiding local optima and enhancing the chances of finding a global solution. Their flexibility and adaptability make them well-suited to handle complex, nonlinear, and multimodal problems like those involved in solar cell parameter estimation (*Sharma et al., 2023b*).

Driven by the No-Free-Lunch theorem (*Adam et al., 2019*), a wide range of new metaheuristic algorithms have been proposed during the last three decades to address global optimization challenges; however, their effectiveness has largely been assessed through standard benchmark functions alone. Thus, it is essential to undertake more in-depth studies to confirm the viability of these optimizers in practical real-world applications. This study, therefore, aims to address a paramount research question identified in the literature: the need for comparative analyses to evaluate the performance of recent state-of-the-art metaheuristics in estimating PV model parameters. To this end, we examine the performance of ten recent metaheuristics in determining PV cell parameters across four case studies, involving the single-diode model (SDM), double-diode model (DDM), triple-diode model (TDM), and PV module model. Figure 1 presents

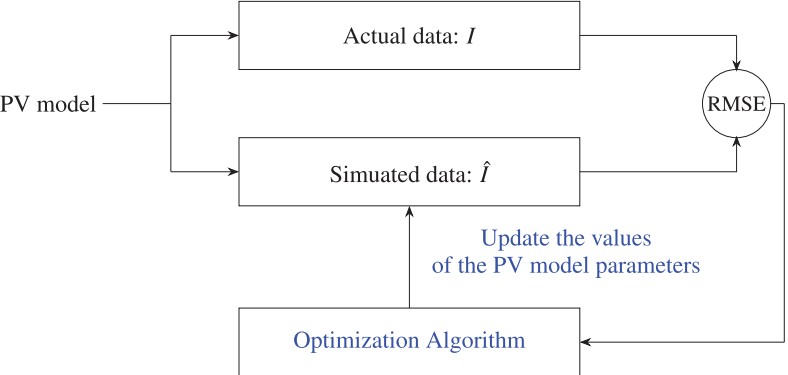

**Figure 1 Flow diagram illustrating the application of metaheuristic algorithms for parameter extraction in PV models.**

a flow diagram illustrating the process of employing metaheuristic algorithms to extract parameters of PV models.

This research article makes several significant contributions to the field of PV model optimization. In this study, we implement ten recent high-performing metaheuristics to tackle the problem of PV parameter estimation for four different solar PV models: SDM, DDM, TDM, and PVMM. The metaheuristic algorithms we have used are the Crayfish Optimization Algorithm (*Jia et al., 2023*), the Golf Optimization Algorithm (*Montazeri et al., 2023*), the Coati Optimization Algorithm (*Dehghani et al., 2023*), the Crested Porcupine Optimizer (*Abdel-Basset, Mohamed & Abouhawwash, 2024*), the Growth Optimizer (*Zhang et al., 2023*), the Artificial Protozoa Optimizer (*Wang et al., 2024*), the Secretary Bird Optimization Algorithm (*Matoušová et al., 2023*), the Mother Optimization Algorithm (*Fu et al., 2024*), the Election Optimizer Algorithm (*Zhou et al., 2024*), and the Technical and Vocational Education and Training-Based Optimizer (*Hubalovska & Major, 2023*). These algorithms have been chosen based on their diversity, popularity in recent studies, and applicability to the PV parameter estimation problem addressed in this research. In addition, it provides a comprehensive evaluation of these algorithms in terms of execution time, number of function evaluations, solutions' optimality, current-voltage (I–V) characteristic curves, power-voltage (P–V) characteristic curves, and convergence rate. The contrastive analysis not only highlights the strengths and weaknesses of each metaheuristic but also identifies the most suitable algorithm for efficient and accurate parameter estimation in PV models. Furthermore, the findings contribute to a deeper understanding of the applicability and performance of metaheuristic algorithms in renewable energy systems.

The article is structured as follows: "PV Models and Mathematical Formulations" describes the considered PV models and presents their mathematical formulations, establishing the foundational concepts necessary for understanding the subsequent sections. "Related Work" offers an overview of some state-of-the-art solutions proposed to address the challenges associated with PV models. "Considered Metaheuristic Algorithms" delves into the selected metaheuristic algorithms, highlighting their working principles and

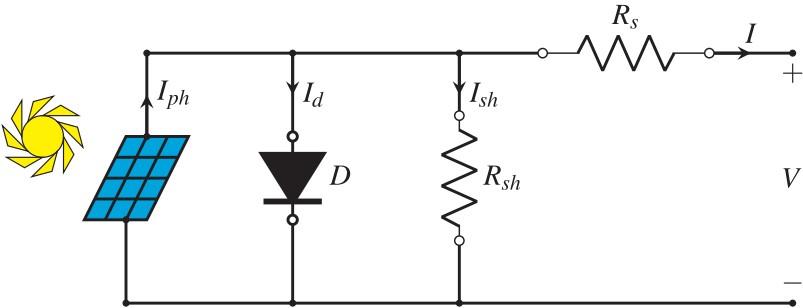

**Figure 2 Diagram of the single-diode model.**

equations, which will be used for contrastive analysis. "Numerical Results and Analysis" presents the numerical results, discussing the performance variations of the considered algorithms, thus providing a detailed comparative assessment. Finally, "Conclusion and Perspectives" summarizes the findings and suggests potential future directions for research in this domain.

## PV MODELS AND MATHEMATICAL FORMULATIONS

PV models are essential components in the study and implementation of solar energy systems. They simulate the behaviour of solar panels in converting sunlight into electricity, helping engineers and researchers optimize the design and efficiency of solar power systems. By accurately predicting the performance of solar panels under various environmental conditions, PV models play a crucial role in the development of sustainable energy solutions. In the following sections, we introduce the most common PV models and present their mathematical formulations (*Petrone, Ramos-Paja & Spagnuolo, 2017*).

### Single-diode model

The single-diode model (SDM) is the most simple and widely used model for simulating the *I–V* (current-voltage) characteristics of a PV cell. It represents the PV cell with a single diode, a shunt resistor, a current source, and a series resistor. This model balances simplicity with accuracy and is often used in software simulations. The relationship between *I–V* of a PV cell in the SDM is presented by Eq. (1), where *I* and *V* are the output current and the output voltage of the PV cell, respectively. Figure 2 is a simplified representation of the equivalent circuit for the SDM (*Petrone, Ramos-Paja & Spagnuolo, 2017*).

$$\begin{cases} \hat{I}(I, V, \mathbf{x}) = I_{ph} - I_{sd}\left(\exp\left(\frac{V + R_s \cdot I}{n \cdot V_t}\right) - 1\right) - \frac{V + R_s \cdot I}{R_{sh}} \\ \mathbf{x} = \left(I_{ph}, I_{sd}, R_s, R_{sh}, n\right) \end{cases} \tag{1}$$

The various symbols employed in Eq. (1) and illustrated in Fig. 2 are explained as follows:

- Photocurrent ($I_{ph}$) is the current produced by the PV cell due to light exposure.
- $I_{sd}$ represents the reverse saturation current of the diodes.

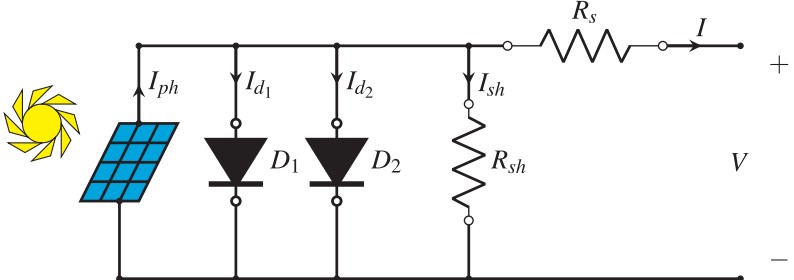

**Figure 3 Diagram of the double-diode model.**

- Series resistance ($R_s$): The resistance due to the current flow through the cell's material and contacts.
- Diode ideality factor ($n$): A factor that accounts for the deviation of the diode from the ideal behaviour, typically ranging from 1 to 2.
- $V_t = \frac{k \cdot T}{q}$ is the thermal voltage, where $T = 273.15 + 33.0$ is the temperature in Kelvin, $k = 1.380649 \times 10^{-23} \, J/K$ is the Boltzmann constant, and $q = 1.602176634 \times 10^{-19} \, C$ is the charge of an electron.
- Shunt resistance ($R_{sh}$): The leakage current paths within the cell.
- $I_d$ is the second term of Eq. (1).
- $I_{sh}$ is the third term of Eq. (1).

## Double-diode model

In the double-diode model (DDM), one additional diode is included that is used for recombination losses in the depletion region of the PV cell. It offers more accuracy than the SDM, especially under low light conditions, but at the cost of increased complexity. The relationship between $I$ and $V$ of a PV cell in the DDM is described by Eq. (2). Figure 3 provides a simplified depiction of the equivalent circuit for the DDM (*Petrone, Ramos-Paja & Spagnuolo, 2017*).

$$\begin{cases} \hat{I}(I, V, \mathbf{x}) = I_{ph} - \sum_{i=1}^{2} \left[ I_{sd_i} \left( \exp\left(\frac{V + R_s \cdot I}{n_i \cdot V_t}\right) - 1 \right) \right] - \frac{V + R_s \cdot I}{R_{sh}} \\ \mathbf{x} = \left( I_{ph}, I_{sd_1}, I_{sd_2}, R_s, R_{sh}, n_1, n_2 \right) \end{cases} \quad (2)$$

In addition to the symbols that have already been explained in Eq. (1), the different symbols used in Eq. (2) and shown in Fig. 3 are explained as follows:

- $I_{sd_1}$ represents the reverse saturation current of the first diode.
- $I_{sd_2}$ represents the reverse saturation current of the second diode.
- Ideality factor of the first diode ($n_1$): Typically close to 1, accounting for diffusion current.
- Ideality factor of the second diode ($n_2$): Typically greater than 2, accounting for recombination current.
- $I_{d_1}$ is the second term of Eq. (2).

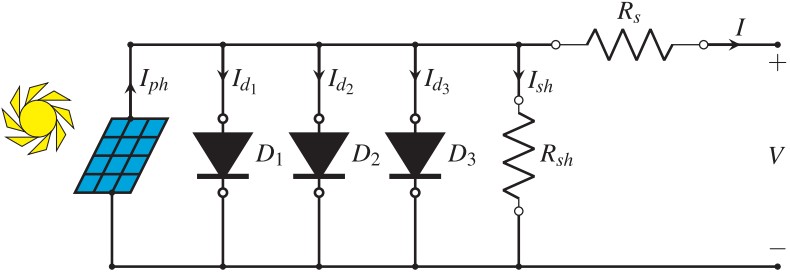

**Figure 4 Diagram of the triple-diode model.**     

- $I_{d_2}$ is the third term of Eq. (2).
- $I_{sh}$ is the fourth term of Eq. (2).

## Triple-diode model

The triple-diode model (TDM) is a more comprehensive representation of a PV cell's behaviour, incorporating three diodes to account for various recombination mechanisms within the cell. This model is particularly useful for high-precision applications and research where the effects of different recombination processes need to be distinguished. The relationship between $I$ and $V$ of a PV cell in the TDM is described by Eq. (3). Figure 4 offers a simplified depiction of the equivalent circuit for the TDM (*Khanna et al., 2015*).

$$\begin{cases} \hat{I}(I, V, \mathbf{x}) = I_{ph} - \sum_{i=1}^{3} \left[ I_{sd_i} \left( \exp\left( \frac{V + R_s \cdot I}{n_i \cdot V_t} \right) - 1 \right) \right] - \frac{V + R_s \cdot I}{R_{sh}} \\ \mathbf{x} = \left( I_{ph}, I_{sd_1}, I_{sd_2}, I_{sd_3}, R_s, R_{sh}, n_1, n_2, n_3 \right) \end{cases} \quad (3)$$

In addition to the symbols already explained in Eqs. (1) and (2), the different symbols used in Eq. (3) and shown in Fig. 4 are:

- $I_{sd_3}$ represents the reverse saturation current of third the diode.
- Ideality factor of the third diode ($n_3$): Typically representing another recombination mechanism.
- $I_{d_1}$ is the second term of Eq. (3).
- $I_{d_2}$ is the third term of Eq. (3).
- $I_{d_3}$ is the fourth term of Eq. (3).
- $I_{sh}$ is the fifth term of Eq. (3).

## PV module model

A PV module model (PVMM) represents the electrical behaviour of a PV module, which consists of multiple PV cells connected in series and/or parallel. This model is crucial for simulating and analyzing the performance of PV modules under various conditions, including changes in sunlight intensity, temperature, and load conditions. By understanding the characteristics of PV modules, engineers can design more efficient solar

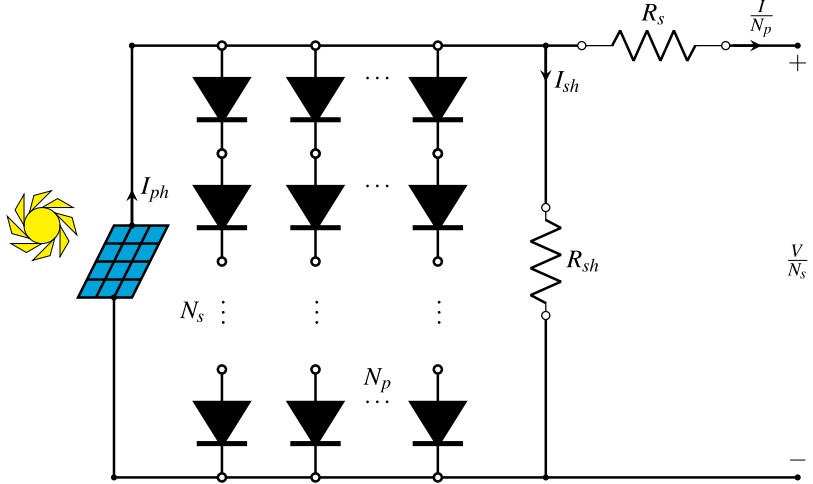

**Figure 5 Diagram of the PV module model.**

energy systems and optimize their performance. The $I–V$ relationship of a PVMM can be described by an extension of the SDM to account for multiple cells. For a module with $N_s$ cells connected in series and $N_p$ strings connected in parallel, Eq. (4) is used. Figure 5 is a simplified representation of the equivalent circuit for a PVMM (*Gu, Xiong & Fu, 2023*).

$$\begin{cases} \hat{I}(I, V, \mathbf{x}) = I_{ph} \cdot N_p - I_{sd} \cdot N_p \left( \exp\left( \frac{V \cdot N_p + R_s \cdot I \cdot N_s}{n \cdot N_p \cdot N_s \cdot V_t} \right) - 1 \right) - \frac{V \cdot N_p + R_s \cdot I \cdot N_s}{R_{sh} \cdot N_s} \\ \mathbf{x} = (I_{ph}, I_{sd}, R_s, R_{sh}, n) \end{cases} \tag{4}$$

## Objective functions

The objective function used in the context of PV models is the root mean square error (RMSE), which is used to optimize the accuracy of the model. The RMSE is calculated by measuring the differences between the actual observed values and the predicted values by the model. The main aim of the RMSE objective function is to minimize the differences between the measured and simulated current values over a range of voltages. This approach ensures that the model accurately represents the real behaviour of the PV cell or module. The smaller the RMSE, the better the model fits the actual performance data. The RMSE expression is given by Eq. (5) (*Khanna et al., 2015*).

$$\text{RMSE} = \sqrt{\sum_{i=1}^{N} \sum_{j=1}^{M} \left( \hat{I}(I_{i,j}, V_{i,j}, \mathbf{x}) - I_{i,j} \right)^2} \tag{5}$$

The RMSE is used to calibrate PV models by adjusting model parameters (*i.e.*, $\mathbf{x}$) to minimize the error between the measured and simulated $I–V$ characteristics. During the parameter extraction process, the objective is to determine the values of the set of parameters that result in the lowest RMSE, indicating the best fit of the model to the observed data. In practical scenarios, the values of $I$ and $V$ are given by the following matrices ($N$ and $M$ are, respectively, the number of rows and columns) (*Easwarakhanthan et al., 1986*).

$$
I = \begin{pmatrix}
-0.2057 & -0.1291 \\
-0.0588 & 0.0057 \\
0.0646 & 0.1185 \\
0.1678 & 0.2132 \\
0.2545 & 0.2924 \\
0.3269 & 0.3585 \\
0.3873 & 0.4137 \\
0.4373 & 0.4590 \\
0.4784 & 0.4960 \\
0.5119 & 0.5265 \\
0.5398 & 0.5521 \\
0.5633 & 0.5736 \\
0.5833 & 0.5900
\end{pmatrix}, \quad
V = \begin{pmatrix}
0.7640 & 0.7620 \\
0.7605 & 0.7605 \\
0.7600 & 0.7590 \\
0.7570 & 0.7570 \\
0.7555 & 0.7540 \\
0.7505 & 0.7465 \\
0.7385 & 0.7280 \\
0.7065 & 0.6755 \\
0.6320 & 0.5730 \\
0.4990 & 0.4130 \\
0.3165 & 0.2120 \\
0.1035 & -0.0100 \\
-0.1230 & -0.2100
\end{pmatrix}
$$

## RELATED WORK

During the last decade, the parameter estimation of PV models has witnessed significant advancements, driven by the increasing demand for accurate and efficient solar energy systems. As reported in *Sharma et al. (2023b)*, researchers have developed a myriad of techniques to enhance the precision of parameter identification, ranging from classical optimization methods to more sophisticated approaches using metaheuristic algorithms. These algorithms aim to enhance the accuracy and efficiency of estimating the unknown parameters in PV models. The following paragraphs review recent contributions in this domain, highlighting their methodologies and outcomes.

The analysis conducted in *Li et al. (2020)* introduced an enhanced adaptive differential evolution (DE) algorithm for estimating the parameters of PV, utilizing a crossover rate sorting mechanism and a dynamic population reduction strategy to improve performance. Similarly, the analysis conducted in *Hao et al. (2020)* presented a multi-strategy success-history-based adaptive DE (SHADE) algorithm, which uses a technique that linearly reduces the population size for the identification of PV model parameters. In this work, a novel weighted mutation operator, Eigen Gaussian random walk strategy and an inferior solution search technique are used to improve the performance of the proposed work. Likewise, the study (*Liang et al., 2020*) developed a self-adaptive ensemble-based DE algorithm to estimate the varying model parameters by combining three different mutation strategies and employing a self-adaptive scheme to balance population diversity and convergence. Additionally, the experiment (*Saadaoui et al., 2021*) proposed a genetic algorithm evaluated on various PV models. To maintain population diversity, a non-uniform mutation operator and an adaptive crossover operator are utilized to get a comprehensive and focused search as the population converges. Likewise, the research (*Kharchouf, Herbazi & Chahboun, 2022*) presented an enhanced DE algorithm for estimating solar PV cell parameters. This algorithm utilized the Lambert W function and a preliminary step to tune the mutation and crossover rates.

The findings detailed in *Abd El-Mageed et al. (2023)* highlighted an improved hybrid optimization algorithm between the queuing search optimization (QSO) algorithm and

DE for PV parameter extraction. This approach enhanced population diversity by applying DE to every solution produced by the QSO. Besides, the work described in *Sharma et al. (2023b)* explored the efficiency of several optimization algorithms for tackling the parameter estimation problem of four PV modules and cells. The analysis of the simulation showed that the wild horse optimizer (*Naruei & Keynia, 2022*) excelled for the Solarex MSX-60 and SS2018 PV modules, while the coot-bird optimization (*Naruei & Keynia, 2021*) algorithm performed best for the LSM20 PV module and the RTC France solar cell. Furthermore, the research outlined in *Memon, Akbari & Zare (2023)* proposed an improved cheetah optimizer, which integrates enhancements to the cheetah optimizer to address parameter estimation in PV models. This improvement was driven by the need for more precise optimization techniques to solve this complex problem. *Bakır (2023)* compared the performance of four methods to enhance the performance of solar PV systems through accurate parameter estimation of solar cells. Experimental results show that the fitness-distance balance-based stochastic fractal search achieved the highest estimation accuracy across different PV models.

The improved moth flame algorithm (*Qaraad et al., 2023*) utilized a local escape technique to maintain the population diversity and boost the algorithm exploration capabilities. Although this algorithm achieved very promising results compared to others, it consumes many number of function evaluations (up to 125,000). Moreover, the hybrid approach (*Janani, Chitti Babu & Krishnasamy, 2023*) combined analytical techniques with the grey wolf algorithm, presenting a solution for the DDM. This combination aimed to leverage the strengths of both methods to achieve better convergence rates and accuracy. Similarly, an integration between the chaos game optimization algorithm and the least squares estimator (*Bogar, 2023*) is proposed to accelerate convergence speed and improve outcomes for PV models. Besides, the squirrel search algorithm (*Maden et al., 2023*) was adapted to estimate the unknown parameters of some PV models by minimizing the root mean squared error between measured and estimated data. This adaptation showed significant improvements in accuracy. Likewise, a new variant of the artificial gorilla troops optimizer is presented in *Abdel-Basset et al. (2023b)* and assessed on parameter estimation for PV models. It uses the ranking-based update strategy and convergence acceleration strategy to improve both exploitation and exploration capabilities. The ranking-based update strategy enhances local and global search abilities, while the convergence acceleration strategy aims to improve global search abilities for quicker solutions. *Sharma et al. (2023a)* introduced teaching learning-based optimization with unique exemplar generation schemes to improve parameter estimation for PV modules, addressing the challenge posed by the model's complexity. By incorporating a modified initialization scheme using chaotic maps and dynamic oppositional learning, along with specialized teacher and learner phases, this hybrid algorithm achieves high precision with low RMSE values across various PV modules.

The Growth Optimizer (*Aribia et al., 2023*) was specifically developed to find the unknown parameters for the KKC and RTC PV modules. However, it required a large number of function evaluations to achieve its results. Furthermore, in the investigation

presented in *Gu et al. (2023)*, a parameter decomposition technique was applied to simplify the complexity of the parameter estimation problem before employing the L-SHADE algorithm. This technique was evaluated using two different PV modules and showed its effectiveness over several competitors. Moreover, the opposition-based initialization particle swarm optimization (*Touabi, Ouadi & Bentarzi, 2023*) improved the original algorithm using opposition-based theory to better solve the parameter identification problem of the SDM. This improvement aimed to enhance the convergence speed and accuracy of the algorithm. In addition, the chimp optimization algorithm (*Yang et al., 2023*) minimized the RMSE between measured and estimated data to find the optimal values of unknown parameters across several PV models. Furthermore, the Harris Hawks optimization algorithm (*Garip, 2023*) was enhanced with fractal maps to propose a new technique, which was used to identify unknown parameters of the RTC France solar cell and PWP PV modules. Moreover, the work investigated in *Chauhan, Vashishtha & Kumar (2023)* introduced an opposition-based learning reptile search algorithm with a Cauchy mutation strategy to improve the identification of parameters in various PV models, enhancing convergence speed and overcoming local minima issues. Lastly, the artificial hummingbird optimizer (*Ayyarao & Kishore, 2024*) was adapted for parameter identification of PV models under three different objective functions, demonstrating its versatility and effectiveness in solving this complex problem.

Recently, several studies have utilized metaheuristic optimization algorithms to tackle the problem of estimating PV model parameters. Various optimization algorithms–including the lion optimizer, arithmetic optimization algorithm, grasshopper optimization algorithm, particle swarm optimization, sine cosine algorithm, salp swarm algorithm, and vortex search algorithm–were compared in *Restrepo-Cuestas & Montano (2024)*. These models have been used to estimate and analyze the parameters of the Bishop model and have been validated using both monocrystalline and polycrystalline photovoltaic cell technologies. Also, *Mohamed et al. (2024)* presented a hybrid optimization algorithm that combines the Kepler optimization algorithm, a ranking-based update mechanism, and exploitation improvement mechanisms. This algorithm demonstrated superior performance for single-, double-, and triple-diode models. In another work, the scholars (*Wu et al., 2024*) proposed an enhanced salp swarm algorithm, referred to as the super-evolutionary Nelder-Mead salp swarm algorithm, for determining the unknown parameters of PV models. The proposed algorithm incorporates a super-evolutionary mechanism with Gaussian-Cauchy mutation and vertical/horizontal crossover to enhance the global optimization and local search capabilities of the standard salp swarm algorithm. Moreover, a combination of an enhanced gas solubility optimization algorithm and a first-order adaptive damping Berndt-Hall-Hall-Hausman method was proposed by *Ramachandran et al. (2024)* to address the theoretical gap in objective function design for PV models. In this study, a new objective function that combines the Berndt-Hall-Hall-Hausman and the Levenberg-Marquardt techniques was introduced to enhance the quality and reliability of estimating the unknown parameters of PV models. Additionally, the authors (*Imade et al., 2024*) proposed an improved JAYA algorithm with Gaussian

mutation and individual weighting factors, which enhances the speed and accuracy of obtaining the values of unknown parameters in PV models. Also, a modified bare-bones imperialist competition algorithm was proposed by *Lei et al. (2024)* to accurately determine the parameters of PV models. In addition, the work in *Ekinci, Izci & Hussien (2024)* developed a novel hybrid optimization algorithm that combines the gazelle optimization algorithm and the Nelder-Mead algorithm to estimate the parameters of PV models. To finish with, a chaos learning butterfly optimization algorithm was proposed for the accurate extraction of PV model parameters (*Ru, 2024*). This algorithm incorporates Cauchy mutation to increase perturbation and help the algorithm escape local optima, a chaos learning strategy to enhance global exploration, local exploitation, and convergence speed/accuracy, and a terminal elimination mechanism to randomly initialize the three worst individual positions to increase population diversity.

These studies demonstrate the effectiveness of metaheuristic approaches in improving PV parameter estimation accuracy and reliability. Table 1 provides a summary of the previously reported metaheuristic-based solutions applied for parameter extraction in PV models. In summary, these advancements highlight the ongoing efforts and innovations in the field of PV parameter estimation, showcasing the diverse approaches and methodologies employed to enhance the accuracy and efficiency of these critical estimations. However, existing metaheuristic optimization techniques for PV model parameter estimation often suffer from at least one of the following drawbacks: falling into local optima, slow convergence speed, and high computational cost.

## CONSIDERED METAHEURISTIC ALGORITHMS

In this section, we provide an overview of the ten metaheuristics used for the contrastive analysis. We will present and briefly explain their pseudocodes, parameters, and equations. Table 2 summarizes the variables utilized in the considered algorithms. It is worth noting that all the metaheuristic algorithms discussed here use Eq. (6) to create the initial population.

$$\begin{cases} \mathbf{x}_i = \mathbf{r}_1 \circ (UB - LB) + LB \\ i \in \{1, \dots, N\} \end{cases} \tag{6}$$

### Crayfish Optimization Algorithm

Inspired by the behaviours of crayfish, a new metaheuristic optimization algorithm, called Crayfish Optimization Algorithm (COA), has been proposed (*Jia et al., 2023*). COA divides these behaviours into three distinct stages to balance exploration and exploitation. The summer resort stage represents the exploration phase where crayfish seek refuge when temperatures are high. The competition stage, which is a part of the exploitation phase, where crayfish compete for the same cave when temperatures are high. The foraging stage, which is another part of the exploitation phase, where crayfish forage for food when the temperature is suitable. Algorithm 1 outlines the working principle of the COA, with each phase illustrated by Eqs. (7) through (12).

**Table 1 Summary of metaheuristic-based solutions for parameter extraction of PV models.**

| Solution | Models | Algorithm(s) used |
|---|---|---|
| *Li et al. (2020)* | SDM, DDM, PVMM | An enhanced adaptive differential evolution algorithm |
| *Hao et al. (2020)* | SDM, DDM, PVMM | A multi-strategy success-history based adaptive differential evolution with linear population size reduction |
| *Liang et al. (2020)* | SDM, DDM, PVMM | A self-adaptive ensemble-based differential evolution algorithm |
| *Saadaoui et al. (2021)* | SDM, DDM, PVMM, ESP-160 PPW PV, STP6-120/36, Photowatt-PWP201 | A genetic algorithm based on non-uniform mutation |
| *Kharchouf, Herbazi & Chahboun (2022)* | SDM, DDM, PVMM | An improved differential evolution algorithm |
| *Abd El-Mageed et al. (2023)* | SDM, DDM, PVMM, TFST40, MCSM55 | An improved queuing search optimization algorithm dependent on the differential evolution |
| *Sharma et al. (2023b)* | R.T.C. France solar cell, LSM20 PV module, Solarex MSX-60 PV module, SS2018P PV module | Spotted hyena optimizer, Sooty tern optimization, Aquila optimization, Harris hawks optimization, Wild horse optimization, Arithmetic optimization algorithm, Atom search optimization, Coot bird optimization |
| *Memon, Akbari & Zare (2023)* | SDM, DDM, PVMM | An Improved cheetah optimizer |
| *Bakır (2023)* | SDM, DDM, PVMM | Fitness-distance balance-based stochastic fractal search, Particle swarm optimization, student psychology-based optimization, adaptive guided differential evolution |
| *Qaraad et al. (2023)* | SDM, Photowatt-PWP 201 model, STM6-40/36 model | An improved moth flame algorithm with local escape operators |
| *Janani, Chitti Babu & Krishnasamy (2023)* | SDM, DDM, PVMM | A hybrid grey wolf optimization |
| *Bogar (2023)* | SDM, DDM, TDM, PVMM | A chaos game optimization-least squares algorithm |
| *Maden et al. (2023)* | SDM, DDM, PVMM, R.T.C. France silicon solar cell, polycrystalline CS6P-220P solar module | Squirrel search algorithm |
| *Abdel-Basset et al. (2023b)* | SDM, DDM, TDM, PVMM | A new variant of the artificial gorilla troops optimizer called ranking-based gorilla troops optimizer |
| *Abdel-Basset et al. (2023a)* | Photowatt-PWP201, Leibold Solar (LSM 20), Leybold Solar (STE 4/100) PV modules | A new variant of teaching learning-based optimization with unique exemplar generation schemes + chaotic maps and dynamic oppositional based learning |
| *Aribia et al. (2023)* | RTC France, Kyocera KC200GT PV modules | Growth optimization |
| *Gu et al. (2023)* | multi-crystalline KC200GT, mono-crystalline SM55 | A success-history adaptation differential evolution with linear population size reduction and decomposition |
| *Touabi, Ouadi & Bentarzi (2023)* | STM6-40/36 module, Photowatt-PWP201 | An opposition based initialization particle swarm optimization |
| *Yang et al. (2023)* | SDM, DDM, TDM, multi-crystalline (KC200GT), polycrystalline (SW255), monocrystalline (SM55) | Chimp optimization algorithm |
| *Garip (2023)* | RTC France photovoltaic cell and Photowatt-PWP 201 photovoltaic models with single, double and three-diodes | A fractional-order Harris Hawks optimization algorithm |
| *Chauhan, Vashishtha & Kumar (2023)* | SDM, DDM, PVMM | An opposition-based learning reptile search algorithm with Cauchy mutation strategy |
| *Ayyarao & Kishore (2024)* | SDM, DDM, PVMM | Artificial hummingbird optimization with three objective functions (RMSE, Lambert W function, Newton–Raphson approach) |

*(Continued)*

| Solution | Models | Algorithm(s) used |
|---|---|---|
| *Restrepo-Cuestas & Montano (2024)* | Bishop model of the monoc-Si cell, Bishop model of the poly c-Si | Ant lion optimizer, arithmetic optimization algorithm, grasshopper optimization algorithm, particle swarm optimization, sine cosine algorithm, salp swarm algorithm, vortex search algorithm |
| *Mohamed et al. (2024)* | SDM, DDM, TDM, RTC France solar cell, Photowatt-PWP201, Ultra 85-P, Ultra 85-P, STP6-120/36, and STM6-40/36 | Kepler optimization algorithm + ranking-based update and exploitation improvement mechanisms |
| *Wu et al. (2024)* | ST40, SM55, C200GT | A super-evolutionary Nelder-Mead salp swarm algorithm |
| *Ramachandran et al. (2024)* | SDM, DDM | An enhanced gas solubility optimization algorithm + first-order adaptive damping Berndt-Hall-Hall-Hausman method |
| *Imade et al. (2024)* | SDM, DDM, TDM, PVMM, monocrystalline SM55, poly-crystalline S75, thin-film ST40 | An improved JAYA algorithm + Gaussian mutation individual weighting factors. |
| *Lei et al. (2024)* | SDM, DDM, PVMM | A modified bare-bones imperialist competition algorithm |
| *Ekinci, Izci & Hussien (2024)* | SDM and DDM of RTC France solar cell Photowatt-PWP201 PV module | Gazelle optimization algorithm combined with the Nelder-Mead algorithm |
| *Ru (2024)* | 12 PV models, YL PV power station model in Guizhou Power Grid | A chaos learning butterfly optimization algorithm |

**Table 2 The list of used parameters and variables.**

| Symbol | Explanation |
|---|---|
| $T_{max}$ | The maximum number of iterations (a scalar value) |
| $t$ | The t-th iteration (a scalar value) |
| $N$ | The population size (a scalar value) |
| $D$ | The dimension of the search space (a scalar value) |
| $X$ | The population of candidate solutions (a matrix of $N$ rows and $D$ columns) |
| $\mathbf{x}_i$ | The i-th candidate solution (a row vector of $D$ columns) |
| $LB$ | The lower boundary values for each decision variable (a row vector of $D$ columns) |
| $UB$ | The upper boundary values for each decision variable (a row vector of $D$ columns) |
| $f(\mathbf{x}_i)$ | The fitness value of the solution $\mathbf{x}_i$ (a scalar value) |
| $\circ$ | The Hadamard product (*i.e.*, the element-wise product) |
| $r_1, r_2, \ldots$ | Random numbers drawn from a uniform distribution between 0 and 1 (scalar values) |
| $\mathbf{r}_1, \mathbf{r}_2, \ldots$ | Row vectors of length $D$, where each element is independently drawn from a uniform distribution between 0 and 1 |
| $\mathbf{g}_1, \mathbf{g}_2, \ldots$ | Row vectors of length $D$, where each element is independently drawn from the Gaussian distribution |
| $\mathbf{I}_S^{(1)}, \mathbf{I}_S^{(2)}, \ldots$ | Row vectors of length $D$, where each element is independently and uniformly drawn from the set $S$ ($S$ contains integers) |
| $\mathbf{L}$ | A Row vectors of length $D$, representing the Lévy flight distribution |

$$\mathbf{y} = \mathbf{x}_i + \left(2 - \left(\frac{t}{T_{max}}\right)\right) \times \mathbf{r}_1 \circ \left(\frac{\mathbf{x}_G + \mathbf{x}_L}{2} - \mathbf{x}_i\right) \tag{7}$$

$$\mathbf{y} = \mathbf{x}_i - \mathbf{x}_{\lfloor r_1 \times (N-1) \rfloor + 1} + \frac{\mathbf{x}_G + \mathbf{x}_L}{2} \tag{8}$$

$$\rho = C_1 \times \left(\frac{1}{\sqrt{2 \times \pi \times \sigma}}\right) \times \exp\left(-\frac{(\tau - \mu)^2}{2 \times \sigma^2}\right) \tag{9}$$

---

**Algorithm 1  Pseudocode of the COA.**

**Input:** $T_{max}$, $N$, $D$, $LB$, $UB$, $f$, $C_1$, $C_2$, $\sigma$, $\mu$

**Output:** The best candidate solution found so far (*i.e.*, $\mathbf{x}^*$)

1  Generate $X$, a random population of candidate solutions, using Eq. (6);

2  $t \leftarrow 1$;

3  **while** $t < T_{max}$ **do**

4      Update $\mathbf{x}_G$ (*i.e.*, the position of the global optimum);

5      **for** $i \leftarrow 1$ **to** $N$ **do**

6          Update $\mathbf{x}_L$ (*i.e.*, the position of the local optimum);

7          $\tau \leftarrow r_1 \times 15 + 20$;

8          **if** $\tau > 30$ **then**

9              **if** $r_2 < 0.5$ **then**

10                 Compute $\mathbf{y}$ using Eq. (7);

11             **else**

12                 Compute $\mathbf{y}$ using Eq. (8);

13             **end**

14         **else**

15             Compute $\rho$ and $Q$ using Eqs. (9) and (10), respectively;

16             **if** $Q > \frac{C_2 + 1}{2}$ **then**

17                 Compute $\mathbf{y}$ using Eq. (11);

18             **else**

19                 Compute $\mathbf{y}$ using Eq. (12);

20             **end**

21         **end**

22         Check and correct the boundary values of $\mathbf{y}$, if necessary;

23         $\mathbf{x}_i \leftarrow \arg\min\{f(\mathbf{x}_i), f(\mathbf{y})\}$;

24     **end**

25     $t \leftarrow t + 1$;

26 **end**

27 Compute the best candidate solution $\mathbf{x}^* \leftarrow \underset{i \in \{1,\dots,N\}}{\arg\min} \{f(\mathbf{x}_i)\}$;

---

$$Q = C_2 \times r_2 \times \frac{f(\mathbf{x}_i)}{f(\mathbf{x}_G)} \tag{10}$$

$$\mathbf{y} = \mathbf{x}_i + \exp\left(-\frac{1}{Q}\right) \times \rho \times (\cos(2 \times \pi \times r_3) - \sin(2 \times \pi \times r_3)) \times \mathbf{x}_G \tag{11}$$

$$\mathbf{y} = \rho \times \left[\left(\mathbf{x}_i - \exp\left(-\frac{1}{Q}\right) \times \mathbf{x}_G\right) + r_4 \times \mathbf{x}_i\right] \tag{12}$$

Zga et al. (2025), *PeerJ Comput. Sci.*, DOI 10.7717/peerj-cs.2646

---

**Algorithm 2**    **Pseudocode of the GOA.**

---

**Input:** $T_{max}, N, D, LB, UB, f$

**Output:** The best candidate solution found so far (*i.e.*, $\mathbf{x}^*$)

1   Generate $X$, a random population of candidate solutions, using Eq. (6);

2   $t \leftarrow 1$;

3   **while** $t < T_{max}$ **do**

4      **for** $i \leftarrow 1$ **to** $N$ **do**

5          Update $\mathbf{x}_G$ (*i.e.*, the position of the global optimum);

6          Compute $\mathbf{y}$ using Eq. (13);

7          Check and correct the boundary values of $\mathbf{y}$, if necessary;

8          $\mathbf{x}_i \leftarrow \mathrm{argmin}\{f(\mathbf{x}_i), f(\mathbf{y})\}$;

9      **end**

10     **for** $i \leftarrow 1$ **to** $N$ **do**

11        Compute $\mathbf{y}$ using Eq. (14);

12        Check and correct the boundary values of $\mathbf{y}$, if necessary;

13        $\mathbf{x}_i \leftarrow \mathrm{argmin}\{f(\mathbf{x}_i), f(\mathbf{y})\}$;

14     **end**

15     $t \leftarrow t + 1$;

16 **end**

17 Compute the best candidate solution $\mathbf{x}^* \leftarrow \underset{i \in \{1,\dots,N\}}{\mathrm{argmin}} \{f(\mathbf{x}_i)\}$;

---

## Golf Optimization Algorithm

A novel game-based metaheuristic optimization technique named the Golf Optimization Algorithm (GOA) (*Montazeri et al., 2023*) is inspired by the strategic dynamics and player behaviours observed in the sport of golf. It is structured into two phases: exploration and exploitation. Algorithm 2 presents the working principle of the GOA, detailing each phase with Eqs. (13) and (14).

$$\mathbf{y} = \mathbf{x}_i + \mathbf{r}_1 \circ \left( \mathbf{x}_G - \mathbf{I}_{\{1,2\}}^{(1)} \circ \mathbf{x}_i \right) \tag{13}$$

$$\mathbf{y} = \mathbf{x}_i + (\mathbf{1}_D - 2 \times \mathbf{r}_1) \circ \frac{\mathbf{r}_2 \circ (UB - LB) + LB}{t} \tag{14}$$

## Coati Optimization Algorithm

The Coati Optimization Algorithm (COA) introduced in *Dehghani et al. (2023)* is a new metaheuristic algorithm that mimics the behaviour of coatis in nature, particularly their strategies in hunting iguanas and escaping predators. The COA algorithm is designed in two phases: exploration and exploitation, representing the coatis' hunting and escaping behaviours, respectively. Algorithm 3 elucidates the working principle of the COA, with each phase detailed through Eqs. (13), (15), and (16).

**Algorithm 3** Pseudocode of the COA.

**Input:** $T_{max}$, $N$, $D$, $LB$, $UB$, $f$

**Output:** The best candidate solution found so far (*i.e.*, $\mathbf{x}^*$)

1   Generate $X$, a random population of candidate solutions, using Eq. (6);

2   $t \leftarrow 1$;

3   **while** $t < T_{max}$ **do**

4      **for** $i \leftarrow 1$ **to** $\lfloor \frac{N}{2} \rfloor$ **do**

5         Update $\mathbf{x}_G$ (*i.e.*, the position of the global optimum);

6         Compute $\mathbf{y}$ using Eq. (13);

7         Check and correct the boundary values of $\mathbf{y}$, if necessary;

8         $\mathbf{x}_i \leftarrow \arg\min\{f(\mathbf{x}_i), f(\mathbf{y})\}$;

9      **end**

10     **for** $i \leftarrow 1 + \lfloor \frac{N}{2} \rfloor$ **to** $N$ **do**

11        Compute $\mathbf{y}$ using Eq. (15);

12        Check and correct the boundary values of $\mathbf{y}$, if necessary;

13        $\mathbf{x}_i \leftarrow \arg\min\{f(\mathbf{x}_i), f(\mathbf{y})\}$;

14     **end**

15     **for** $i \leftarrow 1$ **to** $N$ **do**

16        Compute $\mathbf{y}$ using Eq. (16);

17        Check and correct the boundary values of $\mathbf{y}$, if necessary;

18        $\mathbf{x}_i \leftarrow \arg\min\{f(\mathbf{x}_i), f(\mathbf{y})\}$;

19     **end**

20     $t \leftarrow t + 1$;

21   **end**

22   Compute the best candidate solution $\mathbf{x}^* \leftarrow \underset{i \in \{1,\dots,N\}}{\arg\min} \{f(\mathbf{x}_i)\}$;

$$
\begin{cases}
\mathbf{y} = \begin{cases}
\mathbf{x}_i + \mathbf{r}_1 \circ \left( \mathbf{z} - \mathbf{I}_{\{1,2\}}^{(1)} \circ \mathbf{x}_i \right), & f(\mathbf{z}) < f(\mathbf{x}_i) \\
\mathbf{x}_i + \mathbf{r}_1 \circ (\mathbf{x}_i - \mathbf{z}), & f(\mathbf{z}) \geq f(\mathbf{x}_i)
\end{cases} \\
\mathbf{z} = \mathbf{r}_2 \circ (UB - LB) + LB
\end{cases}
\tag{15}
$$

$$
\mathbf{y} = \mathbf{x}_i + (\mathbf{1}_D - 2 \times \mathbf{r}_1) \circ \left( \mathbf{r}_2 \circ \frac{(UB - LB)}{t} + \frac{LB}{t} \right)
\tag{16}
$$

## Crested Porcupine Optimizer

The research article presented in *Abdel-Basset, Mohamed & Abouhawwash (2024)* introduces a novel metaheuristic algorithm inspired by the defensive behaviours of crested porcupines. This algorithm, named Crested Porcupine Optimizer (CPO), uses four distinct protective mechanisms: sight, sound, odour, and physical attack, each reflecting different

---

**Algorithm 4  Pseudocode of the CPO.**

**Input:** $T_{\max}$, $N$, $D$, $LB$, $UB$, $f$, $T_f$, $\alpha$

**Output:** The best candidate solution found so far (*i.e.*, $\mathbf{x}^*$)

1  Generate $X$, a random population of candidate solutions, using Eq. (6);

2  $t \leftarrow 1$;

3  **while** $t < T_{\max}$ **do**

4    **for** $i \leftarrow 1$ **to** $N$ **do**

5      Update $\mathbf{x}_G$ (*i.e.*, the position of the global optimum);

6      **if** $r_1 < r_2$ **then**

7        **if** $r_3 < r_4$ **then**

8          Compute $\mathbf{y}$ using Eq. (17);

9        **else**

10          Compute $\mathbf{y}$ using Eq. (18);

11        **end**

12      **else**

13        **if** $r_5 < T_f$ **then**

14          Compute $\mathbf{y}$ using Eq. (19);

15        **else**

16          Compute $\mathbf{y}$ using Eq. (20);

17        **end**

18      **end**

19      Check and correct the boundary values of $\mathbf{y}$, if necessary;

20      $\mathbf{x}_i \leftarrow \operatorname{argmin}\{f(\mathbf{x}_i), f(\mathbf{y})\}$;

21    **end**

22    $t \leftarrow t + 1$;

23  **end**

24  Compute the best candidate solution $\mathbf{x}^* \leftarrow \underset{i\in\{1,\dots,N\}}{\operatorname{argmin}}\{f(\mathbf{x}_i)\}$;

---

phases of exploration and exploitation in the optimization process. Algorithm 4 clarifies the operational principle of the CPO, with each stage comprehensively detailed by Eqs. (17)–(20).

$$\mathbf{y} = \mathbf{x}_i + \mathbf{g}_1 \circ \left| 2 \times \mathbf{r}_1 \circ \mathbf{x}_G - \frac{\mathbf{x}_i + \mathbf{x}_{\lfloor r_1 \times (N-1)\rfloor+1}}{2} \right| \tag{17}$$

$$\begin{cases} \mathbf{y} = \left(\mathbf{1}_D - \mathbf{I}_{\{0,1\}}^{(1)}\right) \circ T \\ T = \mathbf{x}_i + \mathbf{I}_{\{0,1\}}^{(1)} \circ \left(\dfrac{\mathbf{x}_i + \mathbf{x}_{\lfloor r_1 \times (N-1)\rfloor+1}}{2} + \mathbf{r}_1 \circ \left(\mathbf{x}_{\lfloor r_2 \times (N-1)\rfloor+1} - \mathbf{x}_{\lfloor r_3 \times (N-1)\rfloor+1}\right)\right) \end{cases} \tag{18}$$

$$\begin{cases} \mathbf{y} = \left(\mathbf{1}_D - \mathbf{I}_{\{0,1\}}^{(1)}\right) \circ \mathbf{x}_i + \mathbf{I}_{\{0,1\}}^{(1)} \circ T \\ T = \left(\mathbf{x}_{\lceil r_1 \times (N-1)\rceil + 1} + S \times \left[\left(\mathbf{x}_{\lceil r_2 \times (N-1)\rceil + 1} - \mathbf{x}_{\lceil r_3 \times (N-1)\rceil + 1}\right) - \delta \times \gamma \times \mathbf{r}_1\right]\right) \\ S = \exp\left(\frac{f(\mathbf{x}_i)}{\sum_{j=1}^{N} f(\mathbf{x}_j) + \varepsilon}\right) \\ \delta = \begin{cases} +1, r_4 \leq 0.5 \\ -1, r_4 > 0.5 \end{cases} \\ \gamma = 2 \times r_5 \times \left(1 - \frac{t}{T_{\max}}\right)^{\frac{1}{T_{\max}}} \end{cases} \tag{19}$$

$$\begin{cases} \mathbf{y} = \mathbf{x}_G + (\alpha \times (\mathbf{1}_D - \mathbf{r}_1) + \mathbf{r}_1) \circ (\delta \times \mathbf{x}_G - \mathbf{x}_i) - \delta \times \gamma \times (\mathbf{r}_2 \circ F) \\ F = \exp(S) \times \left(\mathbf{r}_3 \circ \left(\mathbf{x}_{\lfloor r_1 \times (N-1)\rfloor + 1} - \mathbf{x}_i\right)\right) \end{cases} \tag{20}$$

### Growth Optimizer

A new metaheuristic algorithm called the Growth Optimizer (GO) (*Zhang et al., 2023*), designed to solve both continuous and discrete global optimization problems. The algorithm is inspired by the learning and reflection mechanisms observed in human growth processes. The learning phase involves acquiring knowledge from the outside world (*i.e.*, exploration), while the reflection phase involves self-assessment and adjustment of learning strategies (*i.e.*, exploitation). Algorithm 5 explains the operational principle of the GO, with each stage thoroughly detailed by Eqs. (21) and (22).

$$\begin{cases} G_1 = \mathbf{x}_G - \mathbf{x}_B \\ G_2 = \mathbf{x}_G - \mathbf{x}_W \\ G_3 = \mathbf{x}_B - \mathbf{x}_W \\ G_4 = \mathbf{x}_{\lfloor r_1 \times (N-1)\rfloor + 1} - \mathbf{x}_{\lfloor r_2 \times (N-1)\rfloor + 1} \\ \mathbf{y} = \mathbf{x}_i + \frac{\mathbf{x}_i}{\sum_{j=1}^{N} \mathbf{x}_j} \times \frac{\sum_{j=1}^{4} \|G_j\| \times G_j}{\sum_{j=1}^{4} \|G_j\|} \end{cases} \tag{21}$$

$$\begin{cases} \mathbf{y} = \begin{cases} \begin{cases} \mathbf{r}_1 \circ (UB - LB) + LB, & r_1 < F \\ \mathbf{r}_2 \circ (\mathbf{x}_G - \mathbf{x}_i) + \mathbf{x}_i, & r_1 \geq F \end{cases}, & r_2 < P_2 \\ \mathbf{x}_i, & r_2 \geq P_2 \end{cases} \\ F = 0.01 + 0.99 \times \left(1 - \frac{t}{T_{\max}}\right) \end{cases} \tag{22}$$

### Artificial Protozoa Optimizer

The research article proposed in *Wang et al. (2024)* presents a new bio-inspired metaheuristic algorithm called the Artificial Protozoa Optimizer (APO). This algorithm is inspired by the survival mechanisms of protozoa, including their foraging, dormancy, and reproductive behaviours. The APO is mathematically modeled by Eqs. (23)–(26); and it is implemented in Algorithm 6.

$$\mathbf{y} = \mathbf{r}_1 \circ (UB - LB) + LB \tag{23}$$

$$\mathbf{y} = \mathbf{x}_i + \mathbf{I}_{\{-1,1\}}^{(1)} \circ \mathbf{r}_1 \circ (\mathbf{r}_2 \circ (UB - LB) + LB) \circ \mathbf{I}_{\{0,1\}}^{(2)} \tag{24}$$

---

**Algorithm 5** Pseudocode of the GO.

**Input:** $T_{\max}$, $N$, $D$, $LB$, $UB$, $f$, $P_1$, $P_2$

**Output:** The best candidate solution found so far (*i.e.*, $\mathbf{x}^*$)

1   Generate $X$, a random population of candidate solutions, using Eq. (6);

2   $t \leftarrow 1$;

3   **while** $t < T_{\max}$ **do**

4      **for** $i \leftarrow 1$ **to** $N$ **do**

5         $Z \leftarrow \mathtt{sort}(X)$;

6         $\mathbf{x}_G \leftarrow \mathbf{z}_1$;

7         $\mathbf{x}_B \leftarrow \mathbf{z}_{\lfloor r_1 \times (P_1-2)\rfloor + 2}$;

8         $\mathbf{x}_W \leftarrow \mathbf{z}_{\lfloor r_2 \times (N-P_1-1)\rfloor + P_1 + 1}$;

9         Compute $\mathbf{y}$ using Eq. (21);

10        Check and correct the boundary values of $\mathbf{y}$, if necessary;

11        $\mathbf{x}_i \leftarrow \mathrm{argmin}\{f(\mathbf{x}_i), f(\mathbf{y})\}$;

12      **end**

13      **for** $i \leftarrow 1$ **to** $N$ **do**

14        Update $\mathbf{x}_G$ (*i.e.*, the position of the global optimum);

15        Compute $\mathbf{y}$ using Eq. (22);

16        Check and correct the boundary values of $\mathbf{y}$, if necessary;

17        $\mathbf{x}_i \leftarrow \mathrm{argmin}\{f(\mathbf{x}_i), f(\mathbf{y})\}$;

18      **end**

19      $t \leftarrow t + 1$;

20  **end**

21  Compute the best candidate solution $\mathbf{x}^* \leftarrow \underset{i\in\{1,\dots,N\}}{\mathrm{argmin}}\{f(\mathbf{x}_i)\}$;

---

**Algorithm 6** Pseudocode of the APO.

**Input:** $T_{\max}$, $N$, $D$, $LB$, $UB$, $f$, $p_{\max}$, $\eta$

**Output:** The best candidate solution found so far (*i.e.*, $\mathbf{x}^*$)

1   Generate $X$, a random population of candidate solutions, using Eq. (6);

2   $t \leftarrow 1$;

3   **while** $t < T_{\max}$ **do**

4      $S \leftarrow \{e_q | e_q \in \mathbb{N}, e_q \sim \mathcal{U}(1, N), \forall p \neq q\ e_q \neq e_p, q = 1, \dots, \lceil p_{\max} \times r_1 \times N\rceil\}$;

5      **for** $i \leftarrow 1$ **to** $N$ **do**

6        **if** $i \in S$ **then**

7          **if** $0.5 \times \left(1 + \cos\left(\pi \times \left(1 - \frac{i}{N}\right)\right)\right) > r_2$ **then**

8            Compute $\mathbf{y}$ using Eq. (23);

---

| | **Algorithm 6** (continued) |
|---|---|
| 9 | **else** |
| 10 | Compute **y** using Eq. (24); |
| 11 | **end** |
| 12 | **else** |
| 13 | **if** $0.5 \times \left(1 + \cos\left(\pi \times \frac{t}{T_{\max}}\right)\right) > r_3$ **then** |
| 14 | Compute **y** using Eq. (25); |
| 15 | **else** |
| 16 | Compute **y** using Eq. (26); |
| 17 | **end** |
| 18 | **end** |
| 19 | Check and correct the boundary values of **y**, if necessary; |
| 20 | $\mathbf{x}_i \leftarrow \arg\min\{f(\mathbf{x}_i), f(\mathbf{y})\}$; |
| 21 | **end** |
| 22 | $t \leftarrow t + 1$; |
| 23 **end** | |
| 24 Compute the best candidate solution $\mathbf{x}^* \leftarrow \underset{i \in \{1,\ldots,N\}}{\arg\min}\{f(\mathbf{x}_i)\}$; | |

$$
\begin{cases}
\mathbf{y} = \mathbf{x}_i + \beta \times \left(\mathbf{x}_{\lceil r_1 \times (N-1)\rceil + 1} - \mathbf{x}_i + \frac{1}{\eta}\sum_{k=1}^{\eta} T_k\right) \circ \mathbf{I}_{\{0,1\}}^{(1)} \\
\beta = r_1 \times \left(1 + \cos\left(\pi \times \frac{t}{T_{\max}}\right)\right) \\
T_k = \exp\left(-\left|\frac{f\left(\mathbf{x}_{\lceil r_2 \times i\rceil + 1}\right)}{f\left(\mathbf{x}_{\lceil r_3 \times (N-i-1)\rceil + 1}\right) + \varepsilon}\right|\right)\left(\mathbf{x}_{\lceil r_2 \times i\rceil + 1} - \mathbf{x}_{\lceil r_3 \times (N-i-1)\rceil + 1}\right)
\end{cases}
\tag{25}
$$

$$
\begin{cases}
\mathbf{y} = \mathbf{x}_i + \beta \times \left(\mathbf{x}_n - \mathbf{x}_i + \frac{1}{\eta}\sum_{k=1}^{\eta} T_k\right) \circ \mathbf{I}_{\{0,1\}}^{(1)} \\
\mathbf{x}_n = \left(\left(1 - \frac{t}{T_{\max}}\right) \times \mathbf{I}_{\{-1,1\}}^{(1)} \circ \mathbf{r}_1\right) \circ \mathbf{x}_i \\
T_k = \exp\left(-\left|\frac{f\left(\mathbf{x}_{\lceil r_2 \times (i-k-1)\rceil + 1}\right)}{f\left(\mathbf{x}_{\lceil r_3 \times (i+k-1)\rceil + 1}\right) + \varepsilon}\right|\right)\left(\mathbf{x}_{\lceil r_2 \times (i-k-1)\rceil + 1} - \mathbf{x}_{\lceil r_3 \times (i+k-1)\rceil + 1}\right)
\end{cases}
\tag{26}
$$

## Mother Optimization Algorithm

The Mother Optimization Algorithm (MOA), introduced in *Matoušová et al. (2023)*, is a novel metaheuristic approach inspired by the interactions between a mother and her children. MOA simulates these interactions through three distinct phases: education, advice, and upbringing. The algorithm begins with the education phase, where initial solutions are generated and the search space is explored. In the advice phase, solutions are refined based on guidance principles, enhancing the quality of the search. Finally, the upbringing phase focuses on exploiting the best solutions by making small adjustments to optimize the results. The MOA is implemented in Algorithm 7 and mathematically modeled by Eqs. (27)–(29).

---

| **Algorithm 7** Pseudocode of the MOA. |
|---|

**Input:** $T_{\max}, N, D, LB, UB, f$

**Output:** The best candidate solution found so far (*i.e.*, $\mathbf{x}^*$)

1  Generate $X$, a random population of candidate solutions, using Eq. (6);

2  $t \leftarrow 1$;

3  **while** $t < T_{\max}$ **do**

4      **for** $i \leftarrow 1$ **to** $N$ **do**

5          Update $\mathbf{x}_G$ (*i.e.*, the position of the global optimum);

6          Compute $\mathbf{y}$ using Eq. (27);

7          Check and correct the boundary values of $\mathbf{y}$, if necessary;

8          $\mathbf{x}_i \leftarrow \text{argmin}\{f(\mathbf{x}_i), f(\mathbf{y})\}$;

9      **end**

10     **for** $i \leftarrow 1$ **to** $N$ **do**

11         $Z \leftarrow \text{sort}\,(X)$;

12         $\mathbf{x}_G \leftarrow \mathbf{z}_N$;

13         Compute $\mathbf{y}$ using Eq. (28);

14         Check and correct the boundary values of $\mathbf{y}$, if necessary;

15         $\mathbf{x}_i \leftarrow \text{argmin}\{f(\mathbf{x}_i), f(\mathbf{y})\}$;

16     **end**

17     **for** $i \leftarrow 1$ **to** $N$ **do**

18         Compute $\mathbf{y}$ using Eq. (29);

19         Check and correct the boundary values of $\mathbf{y}$, if necessary;

20         $\mathbf{x}_i \leftarrow \text{argmin}\{f(\mathbf{x}_i), f(\mathbf{y})\}$;

21     **end**

22     $t \leftarrow t + 1$;

23 **end**

24 Compute the best candidate solution $\mathbf{x}^* \leftarrow \underset{i \in \{1,\dots,N\}}{\text{argmin}}\, \{f(\mathbf{x}_i)\}$;

$$y = \mathbf{x}_i + \mathbf{r}_1 \circ (\mathbf{x}_G - 2 \times \mathbf{r}_2 \circ \mathbf{x}_i) \tag{27}$$

$$y = \mathbf{x}_i + \mathbf{r}_1 \circ (\mathbf{x}_i - 2 \times \mathbf{r}_2 \circ \mathbf{z}_N) \tag{28}$$

$$y = \mathbf{x}_i + (\mathbf{1}_D - 2 \times \mathbf{r}_1) \circ \left(\frac{UB - LB}{t}\right) \tag{29}$$

## Secretary Bird Optimization Algorithm

A new population-based optimization algorithm has been proposed by following the behaviour of secretary birds in their natural environment, called the Secretary Bird Optimization Algorithm (SBOA) (*Fu et al., 2024*). The algorithm's design is centred on two primary phases: exploration and exploitation. During the first phase, the SBOA mimics the hunting strategy of secretary birds as they search for prey, which enhances the

---

**Algorithm 8** Pseudocode of the SBOA.

**Input:** $T_{\max}$, $N$, $D$, $LB$, $UB$, $f$

**Output:** The best candidate solution found so far (*i.e.*, $\mathbf{x}^*$)

1 Generate $X$, a random population of candidate solutions, using Eq. (6);

2 $t \leftarrow 1$;

3 **while** $t < T_{\max}$ **do**

4     **for** $i \leftarrow 1$ **to** $N$ **do**

5         Update $\mathbf{x}_G$ (*i.e.*, the position of the global optimum);

6         **if** $t < \frac{T_{\max}}{3}$ **then**

7             Compute $\mathbf{y}$ using Eq. (30);

8         **else if** $\frac{T_{\max}}{3} < t < \frac{2 \times T_{\max}}{3}$ **then**

9             Compute $\mathbf{y}$ using Eq. (31);

10         **else**

11             Compute $\mathbf{y}$ using Eq. (32);

12         **end**

13         Check and correct the boundary values of $\mathbf{y}$, if necessary;

14         $\mathbf{x}_i \leftarrow \operatorname{argmin}\{f(\mathbf{x}_i), f(\mathbf{y})\}$;

15     **end**

16     **for** $i \leftarrow 1$ **to** $N$ **do**

17         Update $\mathbf{x}_G$ (*i.e.*, the position of the global optimum);

18         Compute $\mathbf{y}$ using Eq. (33);

19         Check and correct the boundary values of $\mathbf{y}$, if necessary;

20         $\mathbf{x}_i \leftarrow \operatorname{argmin}\{f(\mathbf{x}_i), f(\mathbf{y})\}$;

21     **end**

22     $t \leftarrow t + 1$;

23 **end**

24 Compute the best candidate solution $\mathbf{x}^* \leftarrow \underset{i \in \{1, \dots, N\}}{\operatorname{argmin}} \{f(\mathbf{x}_i)\}$;

---

algorithm's ability to explore the solution space broadly. In the second phase, the algorithm emulates the birds' escape strategy from predators, focusing on refining the search and converging to optimal solutions. The SBOA is executed in Algorithm 8 and represented by Eqs. (30)–(33).

$$\mathbf{y} = \mathbf{x}_i + \left(\mathbf{x}_{\lfloor r_1 \times (N-1)\rfloor + 1} - \mathbf{x}_{\lfloor r_2 \times (N-1)\rfloor + 1}\right) \circ \mathbf{r}_1 \tag{30}$$

$$\mathbf{y} = \mathbf{x}_G + \exp\left(\left(\frac{t}{T_{\max}}\right)^4\right) \times (\mathbf{r}_1 - 0.5 \times \mathbf{1}_D) \circ (\mathbf{x}_G - \mathbf{x}_i) \tag{31}$$

$$\mathbf{y} = \mathbf{x}_G + \left(1 - \frac{t}{T_{\max}}\right)^{\frac{2 \times t}{T_{\max}}} \times (\mathbf{x}_i \circ 0.5 \times \mathbf{L}) \tag{32}$$

$$\mathbf{y} = \begin{cases} \mathbf{x}_G + \left(1 - \frac{t}{T_{\max}}\right)^2 \times (2 \times \mathbf{r}_1 - \mathbf{1}_D) \circ \mathbf{x}_i, & r_1 < 0.5 \\ \mathbf{x}_i + \mathbf{r}_2 \circ \left(\mathbf{x}_{\lceil r_2 \times (N-1)\rceil + 1} - \mathbf{I}_{\{1,2\}}^{(1)} \circ \mathbf{x}_i\right), & r_1 \geq 0.5 \end{cases} \tag{33}$$

### Election optimizer algorithm

Simulating the democratic electoral process, specifically presidential elections, a new metaheuristic optimization algorithm, called the Election Optimizer algorithm (EOA), has been proposed (*Zhou et al., 2024*). EOA simulates the complete election process, incorporating explicit behaviours observed during elections, such as the nomination of each party and presidential campaigns. Throughout the party nomination phase and to escape local optima, the search space is extended by incorporating varied strategies and innovative suggestions. During the presidential election phase, elite candidates are further refined through televised debates and campaign speeches, maintaining population diversity and enhancing convergence speed. Algorithm 9 summarizes the different steps involved in defining the EOA, and its swarming behaviour is depicted by Eqs. (34)–(37).

$$\mathbf{y} = \mathbf{x}_{\lfloor r_1 \times (N-1)\rfloor + 1} + \alpha \times \left(1 - \frac{t}{T_{\max}}\right) \times \mathbf{r}_1 \circ (\mathbf{x}_i - \mathbf{x}_G) \tag{34}$$

$$\mathbf{y} = \mathbf{r}_1 \circ \mathbf{x}_i + \mathbf{r}_2 \circ (\mathbf{r}_3 \circ (UB - LB) + LB) \tag{35}$$

$$\mathbf{y} = \mathbf{x}_G + \mathbf{r}_1 \circ (\mathbf{x}_G - 2 \times \mathbf{r}_2 \circ \mathbf{x}_i) \tag{36}$$

$$\mathbf{y} = \mathbf{x}_G + (t+1)^{-\frac{2 \times r_1 + 1}{(T_{\max}+1)^2}} \times \mathbf{r}_1 \circ \left(\frac{1}{N}\sum_{i=j}^{N} \mathbf{x}_j - \mathbf{x}_G\right) \tag{37}$$

### Technical and Vocational Education and Training-Based Optimizer

Inspired by the process of teaching work-related skills in training schools and vocational and technical education, a new metaheuristic algorithm has been proposed, known as Technical and Vocational Education and Training-Based Optimizer (TVETBO) (*Hubalovska & Major, 2023*). TVETBO is designed around three main phases: theory and practical education, and individual skills development. In the theory education phase, the algorithm mimics the theoretical knowledge imparted by instructors, enhancing the initial exploration of the search space. During the practical education phase, the algorithm simulates hands-on training, improving solution candidates through iterative refinement. To enhance the exploitation capability of the algorithm, the individual skills development phase focuses on fine-tuning solutions based on personal skill enhancement. Algorithm 10 depicts the pseudocode of the TVETBO, and its updating mechanisms are given by Eqs. (38)–(40).

$$\mathbf{y} = \mathbf{x}_i + \mathbf{r}_1 \circ \left(\mathbf{x}_G - \mathbf{I}_{\{1,2\}}^{(1)} \circ \mathbf{x}_i\right) \tag{38}$$

$$\mathbf{y} = \mathbf{x}_G + \frac{t}{T_{\max}} \times \mathbf{r}_1 \circ (\mathbf{x}_i - \mathbf{x}_G) \tag{39}$$

---

**Algorithm 9   Pseudocode of the EOA.**

**Input:** $T_{\max}$, $N$, $D$, $LB$, $UB$, $f$, $\alpha$

**Output:** The best candidate solution found so far (*i.e.*, $\mathbf{x}^*$)

1  Generate $X$, a random population of candidate solutions, using Eq. (6);

2  $t \leftarrow 1$;

3  **while** $t < T_{\max}$ **do**

4      **for** $i \leftarrow 1$ **to** $N$ **do**

5          Update $\mathbf{x}_G$ (*i.e.*, the position of the global optimum);

6          **if** $t < \frac{T_{\max}}{2}$ **then**

7              **if** $r_1 < 0.5$ **then**

8                  Compute $\mathbf{y}$ using Eq. (34);

9              **else**

10                  Compute $\mathbf{y}$ using Eq. (35);

11              **end**

12          **else**

13              **if** $r_2 < 0.5$ **then**

14                  Compute $\mathbf{y}$ using Eq. (36);

15              **else**

16                  Compute $\mathbf{y}$ using Eq. (37);

17              **end**

18          **end**

19          Check and correct the boundary values of $\mathbf{y}$, if necessary;

20          $\mathbf{x}_i \leftarrow \mathrm{argmin}\{f(\mathbf{x}_i), f(\mathbf{y})\}$;

21      **end**

22      $t \leftarrow t + 1$;

23  **end**

24  Compute the best candidate solution $\mathbf{x}^* \leftarrow \underset{i \in \{1,\dots,N\}}{\mathrm{argmin}} \{f(\mathbf{x}_i)\}$;

---

**Algorithm 10   Pseudocode of the TVETBO.**

**Input:** $T_{\max}$, $N$, $D$, $LB$, $UB$, $f$

**Output:** The best candidate solution found so far (*i.e.*, $\mathbf{x}^*$)

1  Generate $X$, a random population of candidate solutions, using Eq. (6);

2  $t \leftarrow 1$;

3  **while** $t < T_{\max}$ **do**

4      **for** $i \leftarrow 1$ **to** $N$ **do**

5          Update $\mathbf{x}_G$ (*i.e.*, the position of the global optimum);

(Continued)

| | **Algorithm 10** (continued) | |
|---|---|---|
| 6 | | Compute **y** using Eq. (38); |
| 7 | | Check and correct the boundary values of **y**, if necessary; |
| 8 | | $\mathbf{x}_i \leftarrow \arg\min\{f(\mathbf{x}_i), f(\mathbf{y})\}$; |
| 9 | **end** | |
| 10 | **for** $i \leftarrow 1$ **to** $N$ **do** | |
| 11 | | Update $\mathbf{x}_G$ (*i.e.*, the position of the global optimum); |
| 12 | | Compute **y** using Eq. (39); |
| 13 | | Check and correct the boundary values of **y**, if necessary; |
| 14 | | $\mathbf{x}_i \leftarrow \arg\min\{f(\mathbf{x}_i), f(\mathbf{y})\}$; |
| 15 | **end** | |
| 16 | **for** $i \leftarrow 1$ **to** $N$ **do** | |
| 17 | | Compute **y** using Eq. (40); |
| 18 | | Check and correct the boundary values of **y**, if necessary; |
| 19 | | $\mathbf{x}_i \leftarrow \arg\min\{f(\mathbf{x}_i), f(\mathbf{y})\}$; |
| 20 | **end** | |
| 21 | $t \leftarrow t + 1$; | |
| 22 | **end** | |
| 23 | Compute the best candidate solution $\mathbf{x}^* \leftarrow \underset{i \in \{1,\dots,N\}}{\arg\min}\{f(\mathbf{x}_i)\}$; | |

$$\mathbf{y} = \mathbf{x}_i + (\mathbf{1}_D - 2 \times \mathbf{r}_1) \circ \left(\frac{UB - LB}{t}\right) \tag{40}$$

# NUMERICAL RESULTS AND ANALYSIS

This section presents the numerical results and discusses the findings of our comparative analysis. As previously mentioned, the contrastive study encompasses a diverse array of metaheuristic algorithms, including COA (*Jia et al., 2023*), GOA (*Montazeri et al., 2023*), COA (*Dehghani et al., 2023*), CPO (*Abdel-Basset, Mohamed & Abouhawwash, 2024*), GO (*Zhang et al., 2023*), APO (*Wang et al., 2024*), MOA (*Matoušová et al., 2023*), SBOA (*Fu et al., 2024*), EOA (*Zhou et al., 2024*), and TVETBO (*Hubalovska & Major, 2023*). These algorithms were evaluated across four PV models: SDM, DDM, TDM, and PVMM. To distinguish COA (*Jia et al., 2023*) and COA (*Dehghani et al., 2023*), which share the same acronyms, we will refer to the first as COA(1) and the second as COA(2). All the metaheuristic algorithms were executed on a computer equipped with an AMD Ryzen 5 7600 6-Core Processor operating at 3.80 GHz and 16 GB of RAM. The operating system running on this machine is Windows 11 Pro, a 64-bit OS designed for x64-based processors. MATLAB R2019a is the chosen programming language for implementing and running the algorithms, ensuring a robust and efficient computational environment. To evaluate the impact of varying the maximum number of iterations and the population size

**Table 3 The value ranges for decision variables in the PV models.**

| Decision variable | Range |
|---|---|
| $I_{ph}$ | $[0, 1]$ |
| $I_{sd}, I_{sd_1}, I_{sd_2}, I_{sd_3}$ | $[0, 1]$ |
| $R_s$ | $[0, 0.5]$ |
| $R_{sh}$ | $[0, 100]$ |
| $n, n_1, n_2, n_3$ | $[1, 2]$ |

on the performance of the chosen algorithms in solving the four selected PV models, we varied the number of iterations from 1,000 to 5,000 in steps of 1,000 and set the population size to 30, 50, and 100, resulting in 15 possible configurations: $\{C_1, \ldots, C_{15}\}$. The dimensions of the search space for each PV model are 5, 7, 9, and 5 for the SDM, DDM, TDM, and PVMM, respectively. Table 3 summarizes the ranges of permissible values for each design variable. Additionally, Table 4 outlines the parameter settings of the considered metaheuristic algorithms, which were taken from their original articles. Finally, each configuration was run 30 times on each PV model to ensure the validity of the central limit theorem and provide credibility to the statistical tests used to compare performance. For each run, we saved the best solution and its fitness value, the convergence curve, the number of function evaluations, and the runtime.

In order to determine the best configuration for each algorithm: *i.e.*, the best maximum number of iterations and population size, we consider three metrics averaged over 30 runs: the average of the best fitness values, the average of the numbers of function evaluations, and the average of the CPU times. First, we normalize these metrics using min-max normalization to bring them onto a comparable scale. The normalized value for each metric is calculated using the following formula:

$$\hat{v}_k = \frac{v_k - v_{\min}}{v_{\max} - v_{\min}}, \, k \in \{1, \ldots, 30\},$$

where $v_k$ is the original value, $v_{\min}$ is the minimum value, and $v_{\max}$ is the maximum value for the specific metric. Next, we assign weights to each normalized metric based on its importance: $\omega_1$ for the average of the best fitness values, $\omega_2$ for the average of the number of function evaluations, and $\omega_3$ for the average of the runtime. We then compute a composite score for each configuration using the following formula:

$$\begin{cases} S = \omega_1 \times v_1 + \omega_2 \times v_2 + \omega_3 \times v_3 \\ v_1 = \frac{1}{30} \times \sum_{k=1}^{30} \alpha_k \\ v_2 = \frac{1}{30} \times \sum_{k=1}^{30} \beta_k \\ v_3 = \frac{1}{30} \times \sum_{k=1}^{30} \gamma_k \end{cases},$$

where $\alpha_k$, $\beta_k$, and $\gamma_k$ represent the normalized values for the best fitness values, the numbers of function evaluations, and the runtimes, respectively. The values of weights $\omega_1$, $\omega_2$, and $\omega_3$ are set to 0.5, 0.25, and 0.25, respectively. Finally, we rank the configurations based on their composite scores and select the configuration with the lowest composite score as the best one. Tables 1 through 8, located in the Supplemental File, provide a

| Table 4 The settings of the considered metaheuristics. | |
|---|---|
| **Parameter** | **Value** |
| **COA(1)** (*Jia et al., 2023*) | |
| $C_1$ | 0.2 |
| $C_2$ | 3 |
| $\sigma$ | 3 |
| $\mu$ | 2.5 |
| **GOA** (*Montazeri et al., 2023*) | |
| This algorithm operates without any control parameters | |
| **COA(2)** (*Dehghani et al., 2023*) | |
| This algorithm operates without any control parameters | |
| **CPO** (*Abdel-Basset, Mohamed & Abouhawwash, 2024*) | |
| $T_f$ | 0.8 |
| $\alpha$ | 0.2 |
| **GO** (*Zhang et al., 2023*) | |
| $P_1$ | 5 |
| $P_2$ | 0.3 |
| **APO** (*Wang et al., 2024*) | |
| $p_{max}$ | 0.1 |
| $\eta$ | 1 |
| **MOA** (*Matoušová et al., 2023*) | |
| This algorithm operates without any control parameters | |
| **SBOA** (*Fu et al., 2024*) | |
| This algorithm operates without any control parameters | |
| **EOA** (*Zhou et al., 2024*) | |
| $\alpha$ | 2 |
| **TVETBO** (*Hubalovska & Major, 2023*) | |
| This algorithm operates without any control parameters | |

summary of the numerical results obtained for this description. Besides, Table 5 provides a comprehensive summary of the configurations where the selected metaheuristic algorithms achieved the best performance for the considered PV models. These optimal configurations will serve as the basis for the forthcoming comparative analysis. In this table, the rows correspond to the different PV models, while the columns denote the various metaheuristic algorithms. Each cell at the intersection of a row and a column contains a pair of numbers: the first indicates the maximum number of iterations, and the second specifies the population size.

Tables 6–9 present the best solutions found by each metaheuristic algorithm for the SDM, DDM, TDM, and PVMM, and their RMSE values. In addition, Tables 10–13 provide statistical summaries (minimum, maximum, median, mean, and standard deviation) of the best configuration found by the selected metaheuristic algorithms for the same PV models. From Table 6, it can be seen that the algorithms COA(1), CPO, GO, and APO have the best

**Table 5 The best configurations (*i.e.*, maximum number of iterations and population size) for each algorithm.**

|  | COA(1) | GOA | COA(2) | CPO | GO | APO | MOA | SBOA | EOA | TVETBO |
|---|---|---|---|---|---|---|---|---|---|---|
| SDM | (3,000, 50) | (2,000, 100) | (4,000, 100) | (5,000, 30) | (2,000, 300) | (3,000, 100) | (1,000, 100) | (5,000, 100) | (1,000, 100) | (4,000, 30) |
| DDM | (3,000, 50) | (3,000, 50) | (1,000, 100) | (5,000, 50) | (5,000, 30) | (5,000, 100) | (5,000, 30) | (3,000, 100) | (2,000, 100) | (2,000, 30) |
| TDM | (2,000, 50) | (2,000, 50) | (1,000, 50) | (5,000, 50) | (3,000, 30) | (3,000, 50) | (3,000, 100) | (5,000, 100) | (3,000, 50) | (4,000, 30) |
| PVMM | (1,000, 30) | (5,000, 100) | (4,000, 100) | (1,000, 30) | (1,000, 30) | (2,000, 30) | (1,000, 30) | (1,000, 30) | (3,000, 100) | (1,000, 100) |

**Table 6 The best parameters found by each metaheuristic algorithm for the SDM.**

|  | $I_{ph}$ | $I_{sd}$ | $R_s$ | $R_{sh}$ | $n$ | RMSE |
|---|---|---|---|---|---|---|
| COA(1) | 7.6077553000E−01 | 3.2300000000E−07 | 3.6377092300E−02 | 5.3718532775E+01 | 1.4811851558E+00 | 9.8602187789E−04 |
| GOA | 7.6855138090E−01 | 1.3135000000E−06 | 2.5198965400E−02 | 1.4380023543E+01 | 1.6428681410E+00 | 1.0914713470E−02 |
| COA(2) | 7.6249242130E−01 | 1.0307000000E−06 | 3.1157331700E−02 | 6.2411353379E+01 | 1.6084623411E+00 | 2.8564228684E−03 |
| CPO | 7.6079175230E−01 | 3.2370000000E−07 | 3.6363187400E−02 | 5.3553641717E+01 | 1.4813939068E+00 | 9.8614349211E−04 |
| GO | 7.6077553040E−01 | 3.2300000000E−07 | 3.6377092700E−02 | 5.3718521395E+01 | 1.4811851459E+00 | 9.8602187789E−04 |
| APO | 7.6077537770E−01 | 3.2300000000E−07 | 3.6376981400E−02 | 5.3720811160E+01 | 1.4811879595E+00 | 9.8602188389E−04 |
| MOA | 7.6425150120E−01 | 1.1500000000E−08 | 4.6662886200E−02 | 1.9696189392E+01 | 1.2077652016E+00 | 6.1812674132E−03 |
| SBOA | 7.6064498500E−01 | 4.3320000000E−07 | 3.5190098800E−02 | 6.3254552197E+01 | 1.5113337758E+00 | 1.1352185900E−03 |
| EOA | 7.6263824490E−01 | 4.2330000000E−07 | 3.5813121800E−02 | 6.6909162728E+01 | 1.5084601948E+00 | 2.0449632490E−03 |
| TVETBO | 7.3468991870E−01 | 5.7000000000E−09 | 4.9085779100E−02 | 8.5319218618E+00 | 1.1687690533E+00 | 4.1036065111E−02 |

**Table 7 The best parameters found by each metaheuristic algorithm for the DDM.**

|  | $I_{ph}$ | $I_{sd_1}$ | $I_{sd_2}$ | $R_s$ | $R_{sh}$ | $n_1$ | $n_2$ | RMSE |
|---|---|---|---|---|---|---|---|---|
| COA(1) | 7.6077553030E−01 | 3.2300000000E−07 | 0.0000000000E+00 | 3.6377093100E−02 | 5.3718520850E+01 | 1.4811851342E+00 | 1.0529289916E+00 | 9.8602187789E−04 |
| GOA | 7.6510021610E−01 | 3.0000000000E-10 | 3.0900000000E−08 | 5.1236688900E−02 | 1.7479119526E+01 | 1.0187798216E+00 | 1.3646460291E+00 | 7.8737429493E−03 |
| COA(2) | 7.6423345180E−01 | 7.1530000000E−07 | 6.1000000000E−09 | 3.2746584500E−02 | 3.0872072880E+01 | 1.5857077990E+00 | 1.2988787792E+00 | 3.5599768202E−03 |
| CPO | 7.6092528040E−01 | 3.0780000000E−07 | 6.0000000000E-10 | 3.6476930200E−02 | 5.0194853219E+01 | 1.4764003687E+00 | 1.9564443808E+00 | 1.0025597044E−03 |
| GO | 7.6078107890E−01 | 7.4930000000E−07 | 2.2600000000E−07 | 3.6740429600E−02 | 5.5485441965E+01 | 2.0000000000E+00 | 1.4510182297E+00 | 9.8248487610E−04 |
| APO | 7.6079890280E−01 | 2.0150000000E−07 | 2.7250000000E−07 | 3.6491241900E−02 | 5.4150354439E+01 | 1.8453966922E+00 | 1.4676611710E+00 | 9.8478502518E−04 |
| MOA | 7.6458680050E−01 | 3.0000000000E-10 | 0.0000000000E+00 | 5.5126062100E−02 | 1.4022792774E+01 | 1.0000000000E+00 | 1.0000000000E+00 | 1.1816280874E−02 |
| SBOA | 7.6058733160E−01 | 3.9520000000E−07 | 0.0000000000E+00 | 3.5533117500E−02 | 6.0847215232E+01 | 1.5017650714E+00 | 1.9995698908E+00 | 1.0625737349E−03 |
| EOA | 7.6130286360E−01 | 1.4620000000E−07 | 1.8596000000E−06 | 3.4437501100E−02 | 8.1426481879E+01 | 1.4334186824E+00 | 1.8742951004E+00 | 1.7907122489E−03 |
| TVETBO | 7.3059686250E−01 | 0.0000000000E+00 | 1.5631000000E−06 | 1.6727317000E−03 | 3.4662745707E+01 | 1.0000030030E+00 | 1.6709129717E+00 | 3.9436054603E−02 |

RMSE values of approximately 0.000986, indicating the most accurate models among the tested algorithms. From Table 7, it can be observed that the COA(1) emerged as the top performer with the lowest RMSE ($9.86 \times 10^{-4}$), followed closely by the GO and APO, all demonstrating superior accuracy in parameter fitting. Other algorithms like SBOA and CPO also showed good performance with slightly higher RMSE values. Conversely, the MOA and TVETBO have the highest RMSEs, indicating less effective parameter optimization. From Table 8, it is clear that GO, COA(1), and APO demonstrate superior

**Table 8 The best parameters found by each metaheuristic algorithm for the TDM.**

| | $I_{ph}$ | $I_{sd_1}$ | $I_{sd_2}$ | $I_{sd_3}$ | $R_s$ | $R_{sh}$ | $n_1$ | $n_2$ | $n_3$ | RMSE |
|---|---|---|---|---|---|---|---|---|---|---|
| COA(1) | 7.607755030E−01 | 0.000000000E+00 | 0.000000000E+00 | 3.230000000E−07 | 3.637091200E−02 | 5.371853553E+01 | 2.000000000E+00 | 1.000000000E+00 | 1.481185182E+00 | 9.860218778E−04 |
| GOA | 7.623061140E−01 | 1.448000000E−01 | 0.000000000E+00 | 6.220700000E−06 | 7.596951500E−03 | 4.727272641E+01 | 1.898810283E+01 | 1.199695330E+00 | 1.861608395E+00 | 1.883770500E−02 |
| COA(2) | 7.629400611E−01 | 1.945000000E−07 | 1.730000000E−07 | 1.113800000E−06 | 3.112055540E−02 | 9.693435914E+00 | 1.676959815E+01 | 1.959614003E+00 | 1.631292964E+00 | 3.630916766E−03 |
| CPO | 7.610302111E−01 | 3.930000000E−08 | 0.000000000E+00 | 3.215000000E−07 | 3.629974950E−02 | 5.392067073E+01 | 1.768737177E+01 | 1.281563150E+00 | 1.481829182E+00 | 1.004654592E−03 |
| GO | 7.607810789E−01 | 0.000000000E+00 | 7.493000000E−07 | 2.260000000E−07 | 3.674042810E−02 | 5.548543701E+01 | 1.000000000E+01 | 2.000000000E+00 | 1.451018340E+00 | 9.824848761E−04 |
| APO | 7.608532653E−01 | 0.000000000E+00 | 3.996000000E−07 | 2.544000000E−07 | 3.659867410E−02 | 5.377504933E+01 | 1.357597366E+01 | 1.940369660E+00 | 1.461293145E+00 | 9.846328269E−04 |
| MOA | 7.640280820E−01 | 0.000000000E+00 | 0.000000000E+00 | 3.052200000E−06 | 2.484721390E−02 | 8.441188660E+01 | 1.000000000E+00 | 1.000000000E+00 | 1.748961527E+00 | 5.286542951E−03 |
| SBOA | 7.606937087E−01 | 0.000000000E+00 | 4.008000000E−07 | 0.000000000E+00 | 3.550247390E−02 | 6.008595623E+01 | 1.000002418E+00 | 1.503227774E+00 | 1.998511024E+00 | 1.069031359E−03 |
| EOA | 7.617978459E−01 | 4.257000000E−07 | 3.984000000E−07 | 2.190000000E−07 | 3.960724420E−02 | 4.252630879E+01 | 1.656444721E+01 | 1.849948822E+00 | 1.283023314E+00 | 1.430711195E−03 |
| TVETBO | 7.131137843E−01 | 1.280000000E−01 | 3.000000000E−10 | 1.000000000E−10 | 3.045359360E−02 | 1.386017537E+01 | 1.234655382E+01 | 1.159382224E+00 | 1.052929491E+00 | 4.569273051E−02 |

**Table 9 The best parameters found by each metaheuristic algorithm for the PVMM.**

|  | $I_{ph}$ | $I_{sd}$ | $R_s$ | $R_{sh}$ | $n$ | RMSE |
|---|---|---|---|---|---|---|
| COA(1) | 3.8564249370E−01 | 7.1050000000E−06 | 1.1705191000E−02 | 1.0000000000E+02 | 1.0000000000E+00 | 1.2307306856E−02 |
| GOA | 3.8891165400E−01 | 7.0409000000E−06 | 1.3033979300E−02 | 6.4416500874E+01 | 1.0000047656E+00 | 1.5384800935E−02 |
| COA(2) | 3.8754220910E−01 | 1.1082700000E−05 | 6.9116877000E−03 | 7.2597073604E+01 | 1.0436188458E+00 | 1.5336896361E−02 |
| CPO | 3.8564249460E−01 | 7.1050000000E−06 | 1.1705190400E−02 | 1.0000000000E+02 | 1.0000000000E+00 | 1.2307306856E−02 |
| GO | 3.8564249400E−01 | 7.1050000000E−06 | 1.1705191500E−02 | 1.0000000000E+02 | 1.0000000000E+00 | 1.2307306856E−02 |
| APO | 3.8564249370E−01 | 7.1050000000E−06 | 1.1705190900E−02 | 1.0000000000E+02 | 1.0000000000E+00 | 1.2307306856E−02 |
| MOA | 3.8564251360E−01 | 7.1050000000E−06 | 1.1705227700E−02 | 1.0000000000E+02 | 1.0000000000E+00 | 1.2307306856E−02 |
| SBOA | 3.8564249400E−01 | 7.1050000000E−06 | 1.1705191100E−02 | 1.0000000000E+02 | 1.0000000000E+00 | 1.2307306856E−02 |
| EOA | 3.8568606460E−01 | 7.1164000000E−06 | 1.1297964000E−02 | 9.9679917046E+01 | 1.0001382417E+00 | 1.2340266929E−02 |
| TVETBO | 3.7564549540E−01 | 6.5955000000E−06 | 4.0851954000E−03 | 4.7228449322E+01 | 1.0000182179E+00 | 2.7419994376E−02 |

**Table 10 The min, max, median, mean, and std values of the best configuration for the SDM.**

|  | Min | Max | Median | Mean | STD |
|---|---|---|---|---|---|
| COA(1) | 9.8602187789E−04 | 2.2286139909E−01 | 2.2286139909E−01 | 2.1546555318E−01 | 4.0508716351E−02 |
| GOA | 1.0914713470E−02 | 1.5549184840E−01 | 1.1783454059E−01 | 1.0946767129E−01 | 3.7017859012E−02 |
| COA(2) | 2.8564228684E−03 | 6.9849842147E−02 | 1.4020806666E−02 | 1.8177180434E−02 | 1.4502684599E−02 |
| CPO | 9.8614349211E−04 | 8.1731195398E−03 | 3.2889515204E−03 | 3.0046337556E−03 | 1.5749312527E−03 |
| GO | 9.8602187789E−04 | 2.2286139909E−01 | 1.1816280874E−02 | 9.1291948113E−02 | 1.0931804288E−01 |
| APO | 9.8602188389E−04 | 2.2286139909E−01 | 6.1133540563E−03 | 3.4572567552E−02 | 7.5282553190E−02 |
| MOA | 6.1812674132E−03 | 2.2286139909E−01 | 2.2286139909E−01 | 2.1563872803E−01 | 3.9560198628E−02 |
| SBOA | 1.1352185900E−03 | 2.2286139909E−01 | 2.2286139909E−01 | 2.0808019063E−01 | 5.6251691784E−02 |
| EOA | 2.0449632490E−03 | 1.5818955994E−02 | 9.5367020927E−03 | 9.0545666088E−03 | 2.9578909944E−03 |
| TVETBO | 4.1036065111E−02 | 2.5139770042E−01 | 2.1716246888E−01 | 1.9433266834E−01 | 4.6844332197E−02 |

**Table 11 The min, max, median, mean, and std values of the best configuration for the DDM.**

|  | Min | Max | Median | Mean | STD |
|---|---|---|---|---|---|
| COA(1) | 9.8602187789E−04 | 2.2286139909E−01 | 2.2286139909E−01 | 1.9330817906E−01 | 7.6634331785E−02 |
| GOA | 7.8737429493E−03 | 4.5011974031E−01 | 2.1041103296E−01 | 2.3944933750E−01 | 1.0597864377E−01 |
| COA(2) | 3.5599768202E−03 | 1.5276176031E−01 | 3.7858783348E−02 | 5.3869898777E−02 | 4.1271719228E−02 |
| CPO | 1.0025597044E−03 | 5.3959456444E−03 | 1.6446674182E−03 | 2.1573824591E−03 | 1.2001519180E−03 |
| GO | 9.8248487610E−04 | 2.2286139909E−01 | 9.8602187789E−04 | 6.1854099491E−02 | 9.8851700726E−02 |
| APO | 9.8478502518E−04 | 5.6861369797E−03 | 1.0275828979E−03 | 1.4948131129E−03 | 1.0270094600E−03 |
| MOA | 1.1816280874E−02 | 2.2286139909E−01 | 2.2286139909E−01 | 2.0647516923E−01 | 5.1137623103E−02 |
| SBOA | 1.0625737349E−03 | 2.2286139909E−01 | 2.2286139909E−01 | 2.0073039807E−01 | 6.7528130406E−02 |
| EOA | 1.7907122489E−03 | 6.3074169621E−01 | 7.0223576028E−02 | 1.9834809043E−01 | 2.3637996745E−01 |
| TVETBO | 3.9436054603E−02 | 2.7284755418E−01 | 1.9065026597E−01 | 1.8378574345E−01 | 4.9004575210E−02 |

**Table 12 The min, max, median, mean, and std values of the best configuration for the TDM.**

|  | Min | Max | Median | Mean | STD |
|---|---|---|---|---|---|
| COA(1) | 9.8602187789E−04 | 2.2286139909E−01 | 2.2286139909E−01 | 1.7232775411E−01 | 9.3256952572E−02 |
| GOA | 1.8837705005E−02 | 6.0674055824E−01 | 4.1446357323E−01 | 3.9036869495E−01 | 1.5440794026E−01 |
| COA(2) | 3.6309167663E−03 | 2.8528869729E−01 | 1.3968897144E−01 | 1.2406134605E−01 | 7.9780889172E−02 |
| CPO | 1.0046545920E−03 | 4.7983708510E−03 | 1.7045175001E−03 | 2.0671695563E−03 | 1.0722654025E−03 |
| GO | 9.8248487610E−04 | 2.2286139909E−01 | 1.5575126985E−03 | 4.2683968750E−02 | 8.3292624881E−02 |
| APO | 9.8463282698E−04 | 1.1801453894E−02 | 1.6029903814E−03 | 3.1404160184E−03 | 3.1429931120E−03 |
| MOA | 5.2865429516E−03 | 2.2286139909E−01 | 2.2286139909E−01 | 2.1093529828E−01 | 4.6509247156E−02 |
| SBOA | 1.0690313595E−03 | 2.2286139909E−01 | 2.2286139909E−01 | 2.0069028280E−01 | 6.7650414973E−02 |
| EOA | 1.4307111957E−03 | 6.3074169621E−01 | 2.2286141179E−01 | 2.7872543765E−01 | 2.4943253138E−01 |
| TVETBO | 4.5692730515E−02 | 2.4933651766E−01 | 2.2286139911E−01 | 2.0102565366E−01 | 4.3910414758E−02 |

**Table 13 The min, max, median, mean, and std values of the best configuration for the PVMM.**

|  | Min | Max | Median | Mean | STD |
|---|---|---|---|---|---|
| COA(1) | 1.2307306856E−02 | 2.2286139909E−01 | 2.2286139909E−01 | 1.4629386623E−01 | 1.0235437952E−01 |
| GOA | 1.5384800935E−02 | 4.2123499532E−02 | 2.3868033619E−02 | 2.4192423362E−02 | 4.8292250280E−03 |
| COA(2) | 1.5336896361E−02 | 4.2828896260E−02 | 2.4253342306E−02 | 2.4490781021E−02 | 5.9814388442E−03 |
| CPO | 1.2307306856E−02 | 2.0032173551E−02 | 1.4729857469E−02 | 1.5115668237E−02 | 2.2852310769E−03 |
| GO | 1.2307306856E−02 | 2.2286139909E−01 | 1.2307306856E−02 | 4.7399655562E−02 | 7.9810322695E−02 |
| APO | 1.2307306856E−02 | 2.2286139909E−01 | 1.5441204831E−02 | 2.9811714221E−02 | 5.2314594224E−02 |
| MOA | 1.2307306856E−02 | 1.9545166579E−02 | 1.9545166579E−02 | 1.6716218635E−02 | 3.4931096189E−03 |
| SBOA | 1.2307306856E−02 | 2.2286139909E−01 | 2.2286139909E−01 | 1.8793261127E−01 | 7.9438809012E−02 |
| EOA | 1.2340266929E−02 | 1.7468213958E−02 | 1.5619128687E−02 | 1.5486560010E−02 | 1.0173063280E−03 |
| TVETBO | 2.7419994376E−02 | 1.8622600143E−01 | 1.3013899147E−01 | 1.2363965276E−01 | 3.9886423754E−02 |

performance with the lowest RMSE values, indicating high accuracy and reliability. CPO also performs commendably well, slightly trailing behind the top three. Moderate performers such as COA(2), MOA, EOA, and SBOA, while not as precise, still provide reasonable accuracy. However, GOA and TVETBO exhibit considerably higher RMSE values, with TVETBO being the least accurate. These results suggest that GO, COA(1), and APO are the most suitable algorithms for applications requiring high precision in the TDM, whereas GOA and TVETBO may need further optimization to enhance their performance. From Table 9, it is evident that six algorithms (COA(1), CPO, GO, APO, MOA, SBOA) consistently achieved the lowest RMSE values, indicating their superior effectiveness in optimizing the PVMM. In contrast, the EOA algorithm, with a slightly higher RMSE, showed minor variations in parameters, indicating a close but slightly less optimal solution. The TVETBO algorithm, however, diverged significantly, yielding a much higher RMSE and differing parameter values, which points to its relatively lower effectiveness for this specific optimization problem.

Tables 10–13 summarize the evaluation of the ten selected metaheuristic algorithms for the SDM, DDM, TDM, and PVMM, run 30 times, to reveal significant insights into their performance and consistency. From Table 10, it is obvious that CPO and EOA emerge as the most reliable and consistent algorithms, evidenced by their low standard deviations of $1.57 \times 10^{-3}$ and $2.96 \times 10^{-3}$, respectively, and generally low median RMSE values of $3.29 \times 10^{-3}$ and $9.54 \times 10^{-3}$, and mean RMSE values of $3.00 \times 10^{-3}$ and $9.05 \times 10^{-3}$. COA(2) also demonstrates strong performance, balancing low error with a median RMSE of $1.40 \times 10^{-2}$ and a mean RMSE of $1.82 \times 10^{-2}$ with a moderate standard deviation of $1.45 \times 10^{-2}$. However, algorithms like GO, APO, MOA, and SBOA display a wide range of performance, with low minimum RMSE values of $9.86 \times 10^{-4}$ but high variability, indicated by their standard deviations of $1.09 \times 10^{-1}$, $7.53 \times 10^{-2}$, $3.96 \times 10^{-2}$, and $5.63 \times 10^{-2}$, respectively. TVETBO is the least stable and reliable, with high variability (standard deviation of $4.68 \times 10^{-2}$) and generally higher error rates (median RMSE of $2.17 \times 10^{-1}$ and mean RMSE of $1.94 \times 10^{-1}$), suggesting it may need further optimization. From Table 11, it is evident that CPO and APO emerge as the most reliable and consistent algorithms, evidenced by their low standard deviations of $1.20 \times 10^{-3}$ and $1.03 \times 10^{-3}$, respectively, and generally low median RMSE values of $1.64 \times 10^{-3}$ and $1.03 \times 10^{-3}$, and mean RMSE values of $2.16 \times 10^{-3}$ and $1.49 \times 10^{-3}$. COA(2) also demonstrates strong performance, balancing low error with a median RMSE of $3.79 \times 10^{-2}$ and a mean RMSE of $5.39 \times 10^{-2}$ with a moderate standard deviation of $4.13 \times 10^{-2}$. However, algorithms like GO, GOA, MOA, and SBOA display a wide range of performance, with low minimum RMSE values of $9.86 \times 10^{-4}$ but high variability, indicated by their standard deviations of $9.89 \times 10^{-2}$, $1.06 \times 10^{-1}$, $5.11 \times 10^{-2}$, and $6.75 \times 10^{-2}$, respectively. EOA is the least stable and reliable, with high variability (standard deviation of $2.36 \times 10^{-1}$) and generally higher error rates (median RMSE of $7.02 \times 10^{-2}$ and mean RMSE of $1.98 \times 10^{-1}$). From Table 12, it is obvious that CPO and APO emerge again as the most reliable and consistent algorithms, evidenced by their low standard deviations of $1.07 \times 10^{-3}$ and $3.14 \times 10^{-3}$, respectively, and generally low median RMSE values of $1.70 \times 10^{-3}$ and $1.60 \times 10^{-3}$, and mean RMSE values of $2.07 \times 10^{-3}$ and $3.14 \times 10^{-3}$. COA(2) also demonstrates strong performance, balancing low error with a median RMSE of $1.40 \times 10^{-1}$ and a mean RMSE of $1.24 \times 10^{-1}$ with a moderate standard deviation of $7.98 \times 10^{-2}$. However, algorithms like GO, GOA, MOA, and SBOA display a wide range of performance, with low minimum RMSE values of $9.86 \times 10^{-4}$ but high variability, indicated by their standard deviations of $8.33 \times 10^{-2}$, $1.54 \times 10^{-1}$, $4.65 \times 10^{-2}$, and $6.77 \times 10^{-2}$, respectively. EOA is the least stable and reliable, with high variability (standard deviation of $2.49 \times 10^{-1}$) and generally higher error rates (median RMSE of $2.23 \times 10^{-1}$ and mean RMSE of $2.79 \times 10^{-1}$). From Table 13, manifestly, EOA stands out with the smallest mean value of $1.55 \times 10^{-2}$ and the smallest standard deviation of $1.02 \times 10^{-3}$, indicating highly consistent and low-error performance. CPO also demonstrates strong performance with a mean of $1.51 \times 10^{-2}$ and a low standard deviation of $2.29 \times 10^{-3}$, showcasing reliability with minimal variability. MOA shows good consistency with a mean of $1.67 \times 10^{-2}$ and a standard deviation of

**Table 14 Average rankings of the ten algorithms based on Friedman ranking test for SDM.**

|        | COA(1) | GOA   | COA(2) | CPO    | GO     | APO    | MOA    | SBOA   | EOA  | TVETBO |
|--------|--------|-------|--------|--------|--------|--------|--------|--------|------|--------|
| Ranks  | 8.0833 | 5.6333 | 4.0667 | 2.1333 | 4.3333 | 3.3333 | 8.1667 | 7.9167 | 3.4  | 7.9333 |

**Table 15 Average rankings of the ten algorithms based on Friedman ranking test for DDM.**

|        | COA(1) | GOA  | COA(2) | CPO    | GO   | APO    | MOA    | SBOA | EOA  | TVETBO |
|--------|--------|------|--------|--------|------|--------|--------|------|------|--------|
| Ranks  | 7.1333 | 7.7  | 4.4333 | 2.5333 | 3.1  | 1.8333 | 7.5333 | 7.4  | 6.7  | 6.6333 |

**Table 16 Average rankings of the ten algorithms based on Friedman ranking test for TDM.**

|        | COA(1) | GOA    | COA(2) | CPO    | GO     | APO    | MOA   | SBOA | EOA  | TVETBO |
|--------|--------|--------|--------|--------|--------|--------|-------|------|------|--------|
| Ranks  | 5.8167 | 9.2333 | 5.0333 | 2.3667 | 2.6833 | 2.2667 | 6.75  | 6.45 | 7.4  | 7      |

$3.49 \times 10^{-3}$. GOA and COA(2) provide balanced performance with mean values around $2.42 \times 10^{-2}$ and $2.45 \times 10^{-2}$, and standard deviations of $4.83 \times 10^{-3}$ and $5.98 \times 10^{-3}$, respectively, indicating moderate consistency. TVETBO has higher performance with a mean of $1.24 \times 10^{-1}$ and a standard deviation of $3.99 \times 10^{-2}$, though with more variability. COA(1) and SBOA exhibit higher variability and less consistency, with mean values of $1.46 \times 10^{-1}$ and $1.88 \times 10^{-1}$, and standard deviations of $1.02 \times 10^{-1}$ and $7.94 \times 10^{-2}$, respectively. Finally, GO and APO show considerable variability with mean values of $4.74 \times 10^{-2}$ and $2.98 \times 10^{-2}$, and standard deviations of $7.98 \times 10^{-2}$ and $5.23 \times 10^{-2}$, respectively, indicating potential for high results but with less consistent performance.

Tables 14–17 present the average rankings of the ten algorithms based on the Friedman ranking test for SDM, DDM, TDM, and PVMM, respectively. The primary objective of applying this test to each model is to identify the algorithm that excels within the corresponding model. It is important to note that the Friedman ranking test was applied to the RMSE values for each algorithm, which were obtained over 30 runs. Specifically, each table represents a $30 \times 10$ matrix of RMSE values, where the rows correspond to the 30 runs, and the columns correspond to the 10 algorithms. From the analysis of these tables, it is evident that CPO achieved the highest ranking for SDM, APO ranked first for both DDM and TDM, and GO ranked first for PVMM. In addition, the Wilcoxon test was applied to the same data to deepen the analysis, where the highest-ranked algorithm for each model was compared against the remaining algorithms. Specifically, CPO was compared to the other algorithms for SDM, APO was compared to the other algorithms for both DDM and TDM, and GO was compared to the other algorithms for PVMM. Tables 18–21 present these findings. From Table 18, the exact $p$-values and asymptotic $p$-values are all less than 0.05, suggesting that the differences observed between CPO and the other algorithms are statistically significant. This means that for all comparisons in the

**Table 17 Average rankings of the ten algorithms based on Friedman ranking test for PVMM.**

|  | COA(1) | GOA | COA(2) | CPO | GO | APO | MOA | SBOA | EOA | TVETBO |
|---|---|---|---|---|---|---|---|---|---|---|
| Ranks | 6.9167 | 6.5667 | 6.5 | 3.65 | 2.6167 | 3.9667 | 4.15 | 8.3667 | 3.9667 | 8.3 |

**Table 18 Results obtained by the Wilcoxon test for comparing CPO with the remaining algorithms for SDM.**

| Algorithms | $R^+$ | $R^-$ | Exact p-value | Asymptotic p-value |
|---|---|---|---|---|
| CPO vs. COA(1) | 464.0 | 1.0 | 3.726E−9 | 0.000002 |
| CPO vs. GOA | 465.0 | 0.0 | 1.8626E−9 | 0.000002 |
| CPO vs. COA(2) | 461.0 | 4.0 | 1.3038E−8 | 0.000002 |
| CPO vs. GO | 363.0 | 102.0 | 0.006194 | 0.00705 |
| CPO vs. APO | 354.0 | 111.0 | 0.011304 | 0.012096 |
| CPO vs. MOA | 465.0 | 0.0 | 1.8626E−9 | 0.000002 |
| CPO vs. SBOA | 462.0 | 3.0 | 9.314E−9 | 0.000002 |
| CPO vs. EOA | 459.0 | 6.0 | 2.608E−8 | 0.000003 |
| CPO vs. TVETBO | 465.0 | 0.0 | 1.8626E−9 | 0.000002 |

**Table 19 Results obtained by the Wilcoxon test for comparing APO with the remaining algorithms for DDM.**

| Algorithms | $R^+$ | $R^-$ | Exact p-value | Asymptotic p-value |
|---|---|---|---|---|
| APO vs. COA(1) | 462.0 | 3.0 | 9.314E−9 | 0.000002 |
| APO vs. GOA | 465.0 | 0.0 | 1.8626E−9 | 0.000002 |
| APO vs. COA(2) | 465.0 | 0.0 | 1.8626E−9 | 0.000002 |
| APO vs. CPO | 390.0 | 75.0 | 7.296E−4 | 0.001155 |
| APO vs. GO | 295.0 | 170.0 | ≥0.2 | 0.195043 |
| APO vs. MOA | 465.0 | 0.0 | 1.8626E−9 | 0.000002 |
| APO vs. SBOA | 465.0 | 0.0 | 1.8626E−9 | 0.000002 |
| APO vs. EOA | 464.0 | 1.0 | 3.726E−9 | 0.000002 |
| APO vs. TVETBO | 465.0 | 0.0 | 1.8626E−9 | 0.000002 |

table, CPO significantly outperforms the other algorithms in terms of the median performance, with very strong evidence against the null hypothesis of no difference. From Table 19, the exact p-values for all comparisons except for APO vs. GO are very small (often in the range of $10^{-9}$), indicating that the differences are statistically significant. For APO vs. GO, the exact p-value is larger than 0.2, suggesting no statistically significant difference between APO and GO in this case. From Table 20, for most comparisons, the exact p-values are very small (in the range of $10^{-9}$), suggesting that APO significantly outperforms the other algorithms. However, for APO vs. CPO and APO vs. GO, the exact p-values are greater than 0.05, indicating no significant difference in performance. For Table 21, the exact p-values for many comparisons are above 0.05 (like for GO vs. GOA,

**Table 20 Results obtained by the Wilcoxon test for comparing APO with the remaining algorithms for TDM.**

| Algorithms | $R^+$ | $R^-$ | Exact p-value | Asymptotic p-value |
|---|---|---|---|---|
| APO vs. COA(1) | 450.0 | 15.0 | 2.552E−7 | 0.000007 |
| APO vs. GOA | 465.0 | 0.0 | 1.8626E−9 | 0.000002 |
| APO vs. COA(2) | 465.0 | 0.0 | 1.8626E−9 | 0.000002 |
| APO vs. CPO | 194.0 | 271.0 | ≥0.2 | 1 |
| APO vs. GO | 303.0 | 162.0 | 0.15188 | 0.144193 |
| APO vs. MOA | 464.0 | 1.0 | 3.726E−9 | 0.000002 |
| APO vs. SBOA | 462.0 | 3.0 | 9.314E−9 | 0.000002 |
| APO vs. EOA | 464.0 | 1.0 | 3.726E−9 | 0.000002 |
| APO vs. TVETBO | 465.0 | 0.0 | 1.8626E−9 | 0.000002 |

**Table 21 Results obtained by the Wilcoxon test for comparing GO with the remaining algorithms for PVMM.**

| Algorithms | $R^+$ | $R^-$ | Exact p-value | Asymptotic p-value |
|---|---|---|---|---|
| GO vs. COA(1) | 420.0 | 45.0 | 3.05E−5 | 1 |
| GO vs. GOA | 325.0 | 140.0 | 0.05768 | 0.055767 |
| GO vs. COA(2) | 325.0 | 140.0 | 0.05768 | 0.055767 |
| GO vs. CPO | 325.0 | 140.0 | 0.05768 | 0.055767 |
| GO vs. APO | 304.5 | 130.5 | 0.06075 | 0.055785 |
| GO vs. MOA | 295.0 | 140.0 | 0.09634 | 1 |
| GO vs. SBOA | 422.0 | 13.0 | 3.278E−7 | 1 |
| GO vs. EOA | 325.0 | 140.0 | 0.05768 | 0.055767 |
| GO vs. TVETBO | 402.0 | 63.0 | 2.316E−4 | 0.000471 |

GO vs. COA(2), *etc.*), indicating no significant difference between GO and the other algorithms in those cases. However, for some comparisons (like GO vs. SBOA and GO vs. TVETBO), the exact p-values are very small, indicating statistical significance in favour of GO.

First, Fig. 6A demonstrates that CPO and EOA achieve the most desirable solution distributions, marked by low variability and tight clustering of data points around the median, indicating high consistency and minimal outliers. In contrast, GOA, GO, and TVETBO exhibit high variability, with GOA and TVETBO particularly yielding lower-quality solutions. Although GO presents some solutions near the performance level of CPO and EOA, TVETBO's results are marked by numerous outliers, suggesting unstable behaviour. Similarly, COA(2) and APO approximate the performance of CPO and EOA, though to a slightly lesser extent. The remaining algorithms, however, show inferior performance, accompanied by significant outliers. Next, in Fig. 6B, CPO and APO maintain their consistent behaviour, with minimal variability in their solution distributions. However, GOA, COA, GO, EOA, and TVETBO display substantial

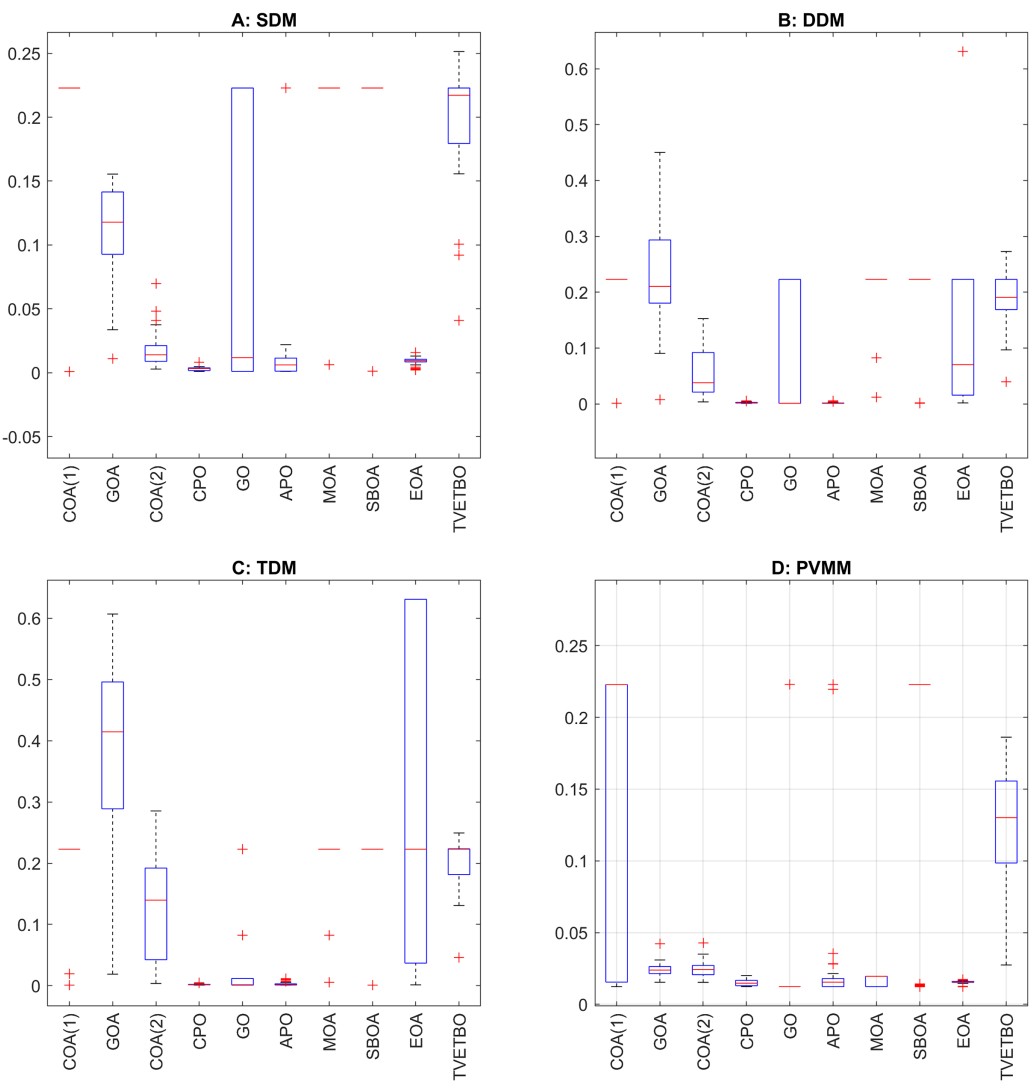

**Figure 6** (A–D) Boxplot of the best fitness values obtained by COA(1), GOA, COA(2), CPO, GO, APO, MOA, SBOA, EOA, and TVETBO.

variability, indicating a degree of instability in their performance. Despite this instability, GO demonstrates competitive performance levels comparable to CPO and APO. In contrast, the remaining algorithms perform poorly, exhibiting numerous outliers and failing to achieve reliable outcomes. Then, Fig. 6C depicts that CPO, GO, and APO exhibit the most favourable variability, though GO contains some outliers. In contrast, GOA, COA (2), EOA, and TVETBO show high levels of variability, indicating inconsistent performance. The remaining algorithms consistently demonstrate poor performance, with numerous outliers reflecting their lack of reliability. Finally, in Fig. 6D, GO surprisingly achieves the best outcome, with only a single outlier. It is closely followed by GOA, COA (2), CPO, APO, MOA, SBOA, and EOA, all of which display similar and strong performance levels. In contrast, COA(1) and TVETBO perform poorly, exhibiting high variability and lower reliability in their results.

Figures 7 and 8 illustrate the comparative performance of the considered metaheuristic algorithms in terms of runtime and the number of function evaluations. Figure 7A, depicting the runtime of the SDM, shows that APO is the fastest algorithm, completing its computations in approximately 20 s, while SBOA is the slowest, taking around 280 s. This significant difference in runtime suggests that APO is highly efficient, whereas SBOA may involve more extensive computations. Figure 8A, which details the number of function evaluations of the SDM, indicates that APO also performs the fewest evaluations, roughly 0.5 million, in contrast to SBOA's highest count of around 6.5 million. These results imply a strong correlation between the number of evaluations and runtime, as algorithms, such as SBOA, which perform more evaluations, naturally require more time. On the other hand, algorithms (COA(1), GO, and EOA) show a balance with moderate runtime and function evaluations, suggesting they are efficient yet thorough in their optimization process. The trade-off between computational time and optimization accuracy is evident, as some algorithms opt for speed and fewer evaluations, while others, like MOA and SBOA, invest more time and computational effort to potentially achieve more precise solutions. From Fig. 8B, SBOA exhibits the highest number of function evaluations, surpassing 3.5 million, suggesting a thorough search process but at the cost of computational resources. In contrast, APO and TVETBO show the lowest function evaluations (FEs), indicating potentially faster but less exhaustive searches. Figure 7B highlights the runtime in seconds, where SBOA again stands out with the longest runtime of approximately 180 s, consistent with its high FEs. CPO and GO also have considerable runtime, indicating intensive computations. On the other hand, APO and TVETBO demonstrate significantly shorter runtimes, around 20 and 50 s respectively, aligning with their lower FEs. Figure 7C displays the runtime in seconds for each algorithm. Notably, MOA and SBOA have the highest execution times, both exceeding 250 s, indicating these algorithms are computationally intensive. In contrast, APO exhibits the shortest runtime, taking less than 10 s. COA(1), COA(2), and EOA also show relatively low execution times, under 50 s. CPO and GO fall in the mid-range, with runtimes around 100 and 60 s, respectively. TVETBO has a moderate execution time of approximately 75 s. Figure 8C shows the number of FEs required by each algorithm. MOA and SBOA again rank highest, each requiring close to $6 \times 10^7$ evaluations, indicating these algorithms perform extensive searches to find solutions. CPO and GO require fewer evaluations, around $2 \times 10^7$ and $1.2 \times 10^7$ respectively. APO requires the fewest evaluations, just below $2 \times 10^6$, aligning with its short runtime. COA(1), COA(2), and EOA have relatively low evaluation counts, similar to their shorter runtimes. TVETBO also has a moderate number of evaluations, approximately $7 \times 10^6$. From Fig. 7D, COA(1), CPO, GO, APO, MOA, and SBOA exhibit very low runtimes, indicating that these algorithms are highly computationally efficient. They can quickly arrive at solutions, which is beneficial for time-sensitive applications. In contrast, GOA, COA(2), and EOA have significantly higher runtimes, with GOA being the most time-consuming at around 220 seconds, followed by EOA at approximately 120 s and COA(2) at around 100 s. This suggests that while these algorithms may offer good performance, they require much more time to compute their solutions. TVETBO also

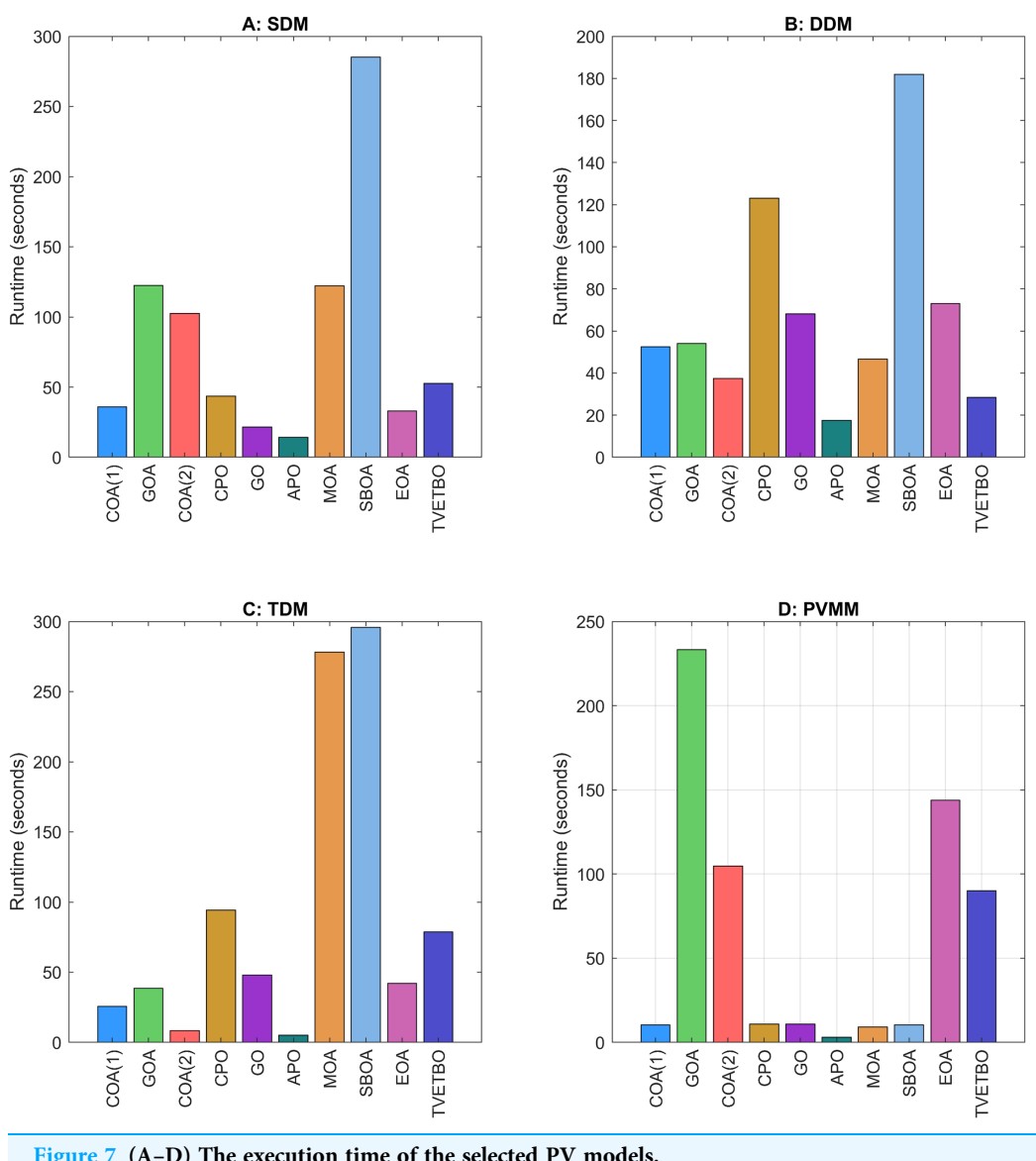

**Figure 7** (A–D) The execution time of the selected PV models.

shows a moderately high runtime of around 80 s, placing it in a middle ground between the very efficient algorithms and the more time-intensive ones. From Fig. 8D, GOA requires the highest number of function evaluations, nearly $6 \times 10^7$, indicating a high computational cost in terms of the number of iterations needed to reach a solution. COA(2) also requires a substantial number of evaluations, around $3 \times 10^7$, followed by EOA and TVETBO at approximately $2 \times 10^7$ each. This high number of evaluations corresponds with their longer runtimes, suggesting a direct relationship between the number of evaluations and the computational time required. On the other hand, COA(1), CPO, GO, APO, MOA, and SBOA have significantly lower numbers of function evaluations, indicating these algorithms can reach solutions more quickly and with fewer computational steps, enhancing their efficiency. This lower computational demand makes

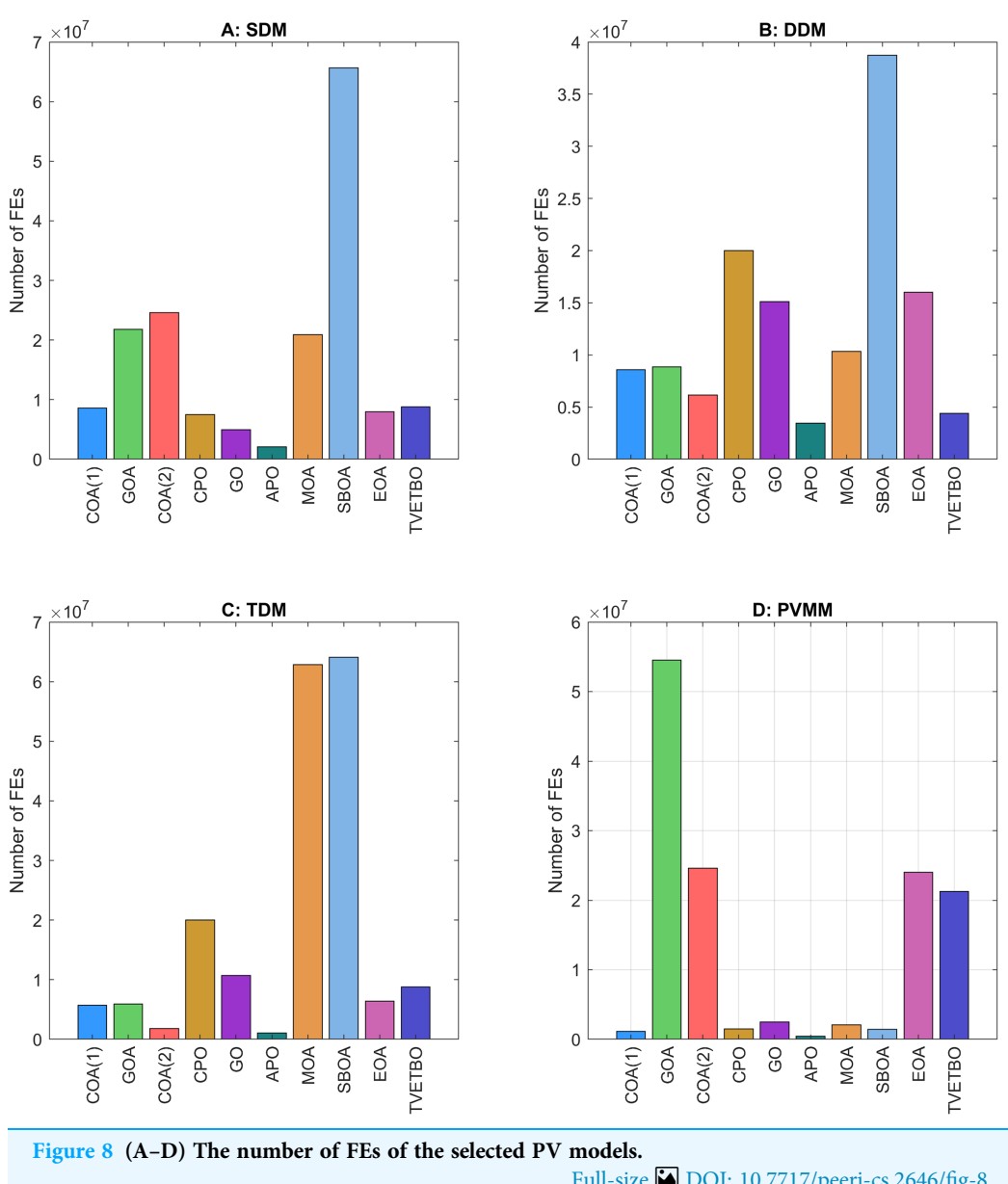

**Figure 8** (A–D) The number of FEs of the selected PV models.

these algorithms particularly suitable for applications where computational resources are limited or where rapid solutions are required.

Figure 9 shows the convergence curves of the selected metaheuristics for the considered PV models, each showing the RMSE as a function of the number of iterations. In Fig. 9A, COA(1), GOA, CPO, and APO exhibit rapid convergence, quickly reducing RMSE to around $10^{-2}$ within the first few hundred iterations, demonstrating their efficiency in reaching near-optimal solutions swiftly. COA(2), MOA, and SBOA show a more gradual but steady convergence, with RMSE stabilizing below $10^{-2}$ after a larger number of iterations, indicating a consistent approach to optimization. EOA and COA(2) feature stepped convergence patterns with significant periodic improvements, reflecting a more phased optimization strategy. In contrast, TVETBO displays slower convergence,

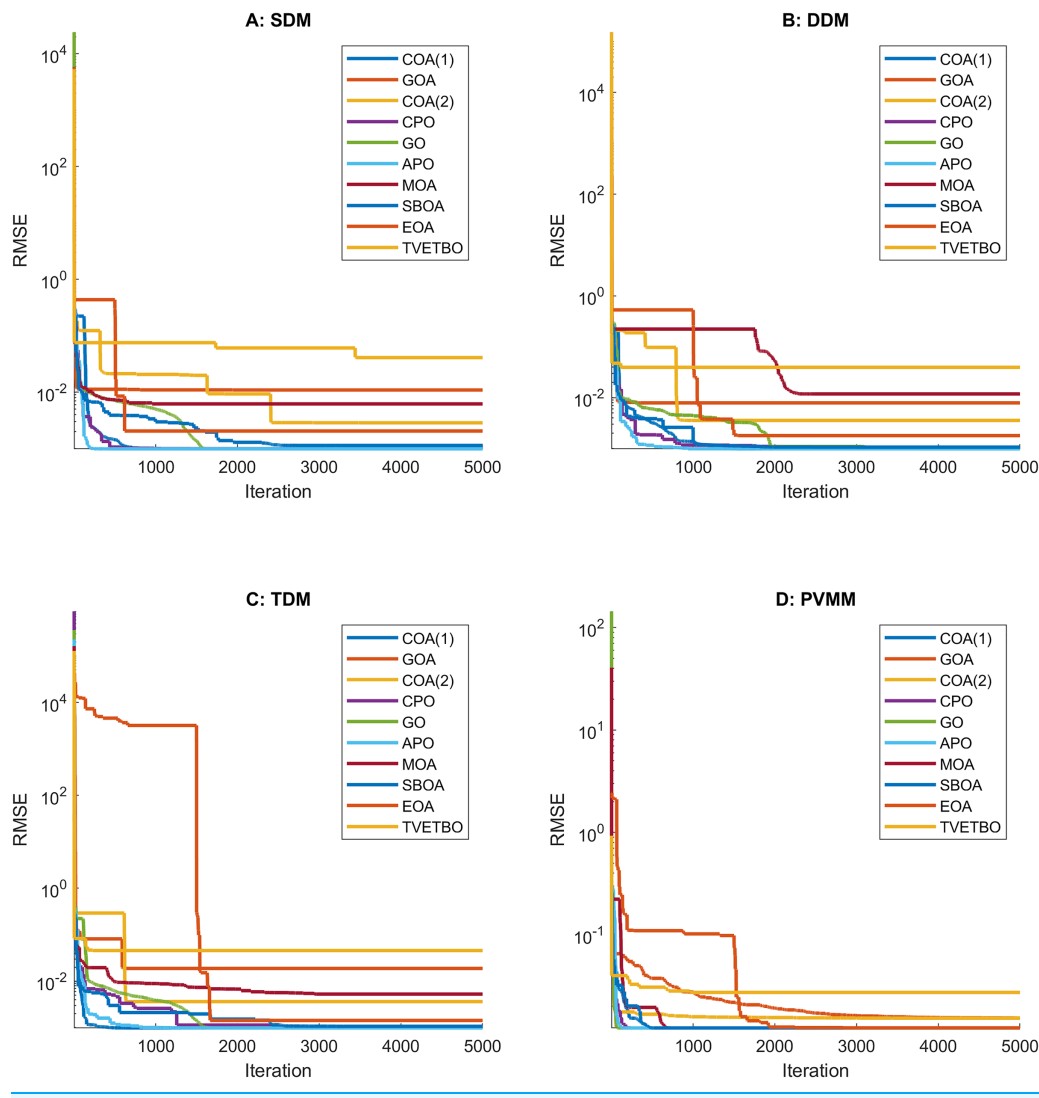

**Figure 9** (A–D) Convergence curves of the best fitness values obtained by COA(1), GOA, COA(2), CPO, GO, APO, MOA, SBOA, EOA, and TVETBO.

stabilizing at a higher RMSE value around $10^{-1}$, suggesting less efficiency in optimization compared to the others. In Fig. 9B, COA(1), GOA, and APO show rapid initial decreases in RMSE, with GOA and APO stabilizing quickly, indicating efficient convergence to optimal solutions. COA(2) and SBOA exhibit step-wise improvements, suggesting periodic significant enhancements in solution quality. CPO, GO, and EOA display more gradual reductions, with CPO and GO achieving steady convergence, while EOA shows a late but drastic improvement phase. MOA demonstrates a delayed yet effective convergence around the 2000th iteration. Conversely, TVETBO converges quickly but stabilizes at a higher RMSE, indicating potentially less accurate results. In Fig. 9C, COA(1), APO, and MOA demonstrate rapid convergence, stabilizing around an RMSE of $10^{-2}$ within the first 200 iterations, indicating efficient early optimization. GOA and COA(2) exhibit two-phase convergence, with steep initial drops followed by further refinement around 500–600

iterations. CPO and SBOA show steady and persistent RMSE reduction, achieving stability around $10^{-2}$ after more extended periods (around 2,000 iterations). GO follows a similar pattern but stabilizes slightly earlier at around 1,500 iterations. EOA is unique, with minimal improvements initially and a sharp RMSE decrease around 1,600 iterations, indicating delayed yet effective optimization. TVETBO converges rapidly within the first 200 iterations but stabilizes slightly above $10^{-2}$, suggesting quicker but less thorough convergence. In Fig. 9D, COA(1), COA(2), CPO, GO, and SBOA show rapid initial convergence, achieving stable RMSE values quickly, indicating efficient performance. GOA and EOA demonstrate slower, more gradual convergence, requiring more iterations to achieve lower RMSE values, which may suggest they are more thorough in exploring the solution space. APO, MOA, and TVETBO show balanced performance with steady convergence, achieving stable solutions within a moderate number of iterations.

Figure 10 the current-voltage curves of the selected metaheuristics for the considered PV models, where each plot compares actual data (blue lines) with simulated data (red triangles). In Fig. 10A, most algorithms, including COA(1), GOA, COA(2), CPO, GO, APO, MOA, SBOA, and EOA, show an excellent fit, with the simulated data closely following the actual data, indicating high accuracy in modelling the PV cell's performance. These curves reveal that these algorithms effectively capture the characteristics of the photovoltaic cells, ensuring precise simulations. On the other hand, the TVETBO algorithm, while generally providing a good fit, shows a slight deviation between the simulated and actual data near the knee of the curve, suggesting it may be slightly less accurate than the others. In Fig. 10B, all algorithms, including COA(1), GOA, COA(2), CPO, GO, APO, MOA, SBOA, EOA, and TVETBO, show a close match between the simulated and actual data, indicating their effectiveness in parameter optimization for the DDM. Notably, the curves for COA(1), GOA, and APO exhibit nearly perfect alignment, reflecting their superior performance in minimizing the RMSE and accurately capturing the PV characteristics. Algorithms like MOA and TVETBO also perform well but show slightly larger deviations in certain regions, suggesting minor discrepancies in their optimization results. In Fig. 10C, COA(1), GOA, COA(2), CPO, GO, APO, MOA, SBOA, and EOA all show excellent alignment between the actual and simulated data, indicating high accuracy in parameter estimation and model representation. TVETBO also performs well, although it shows a slight deviation at higher voltages, suggesting that while it is generally effective, it may require further refinement to achieve the same level of precision as the other algorithms. In Fig. 10D, all algorithms show a moderate level of accuracy, with the simulated data closely aligning with the actual data. This loose alignment indicates how each algorithm replicates the real-world behaviour of the PVMM. Among these, COA(1), COA(2), MOA, and SBOA stand out for their higher precise modelling, as evidenced by the near-perfect match between simulated and actual data points.

Figure 11 illustrates the power-voltage curves of the selected metaheuristics for the considered PV models, where each plot compares actual data (blue lines) with simulated data (red triangles). In Fig. 11A, most algorithms, including COA(1), GOA, COA(2), CPO, GO, APO, MOA, SBOA, and EOA, exhibit an excellent fit between the simulated and

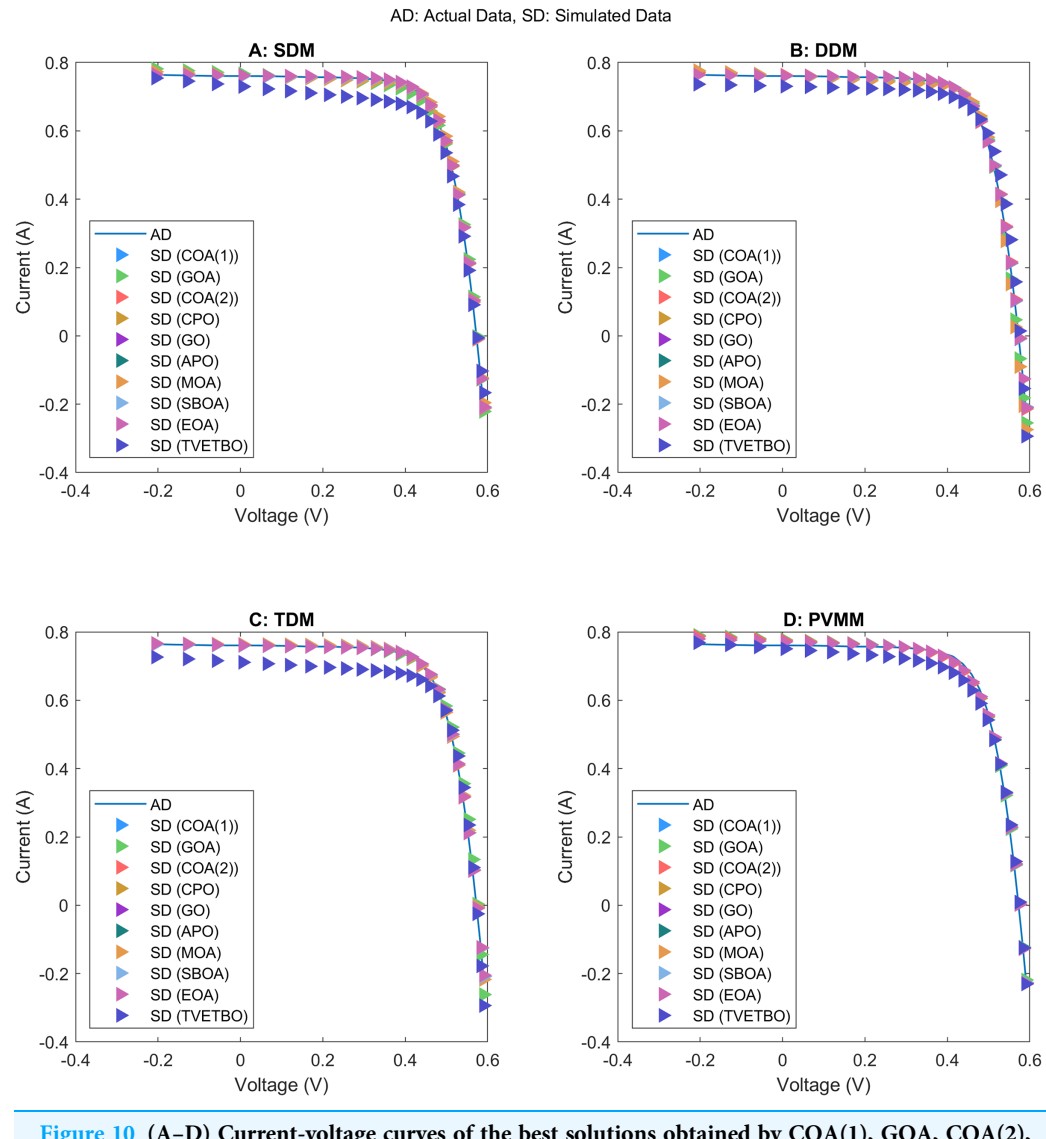

**Figure 10** (A–D) Current-voltage curves of the best solutions obtained by COA(1), GOA, COA(2), CPO, GO, APO, MOA, SBOA, EOA, and TVETBO.

actual data, with simulated points closely following the actual data across the entire voltage range. This indicates high accuracy in representing the power characteristics of the PV cells. On the other hand, the TVETBO algorithm shows a generally good fit but deviates slightly near the peak power point, suggesting it may be less precise than the other algorithms. In Fig. 11B, all algorithms, including COA(1), GOA, COA(2), CPO, GO, APO, MOA, SBOA, EOA, and TVETBO, show a high degree of accuracy in replicating the actual data, with the power curves closely following the actual P-V characteristics. Notably, COA (1), GOA, and APO show near-perfect alignment with the actual data, indicating their superior performance in minimizing errors and capturing the PV system's behaviour. Algorithms such as MOA and TVETBO, while still accurate, exhibit slightly larger deviations near the peak power point, suggesting minor discrepancies in their optimization

AD: Actual Data, SD: Simulated Data

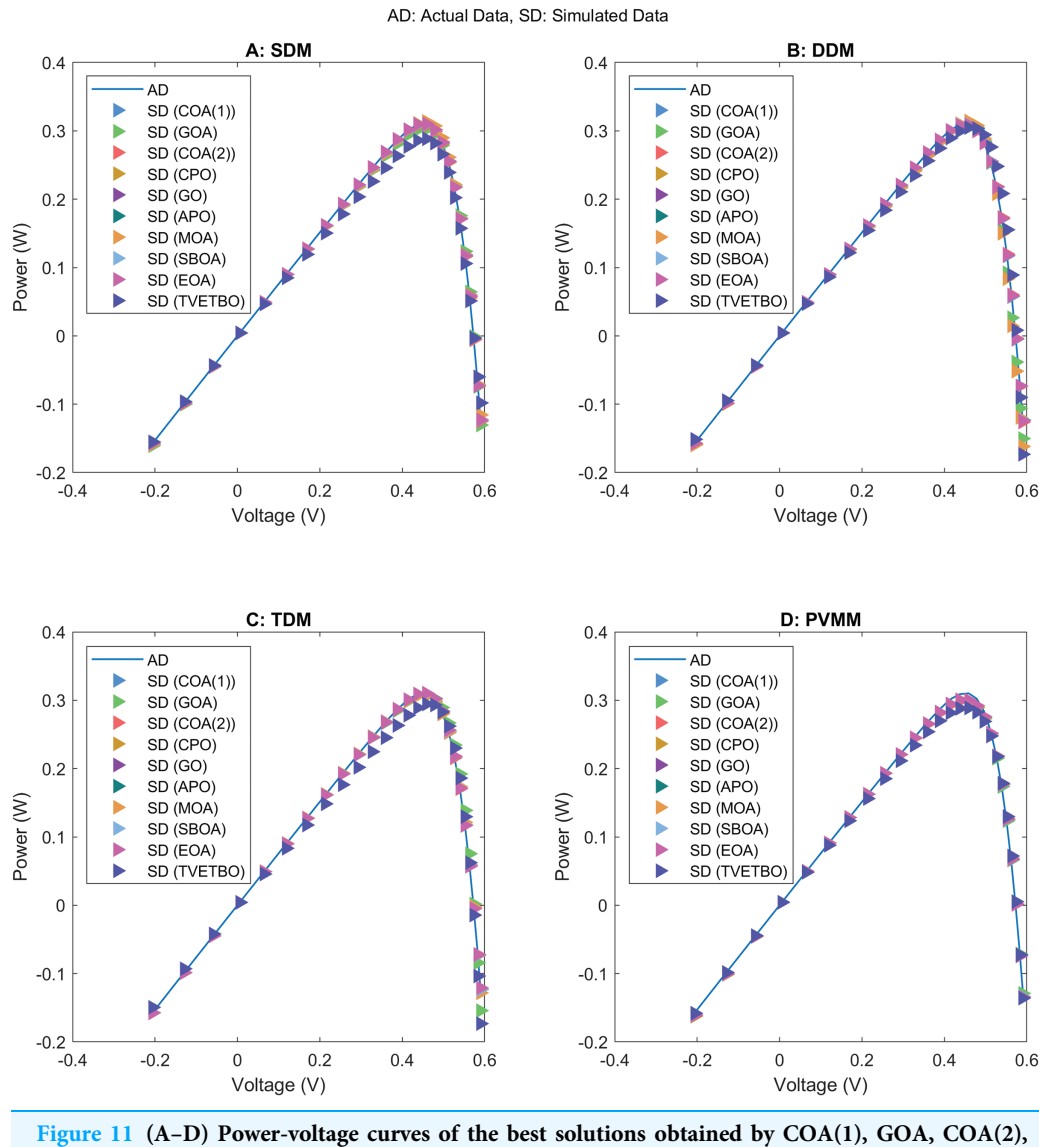

**Figure 11** (A–D) Power-voltage curves of the best solutions obtained by COA(1), GOA, COA(2), CPO, GO, APO, MOA, SBOA, EOA, and TVETBO.

results. In Fig. 11C, COA(1), GOA, COA(2), CPO, GO, APO, MOA, SBOA, and EOA show a strong correlation between the actual and simulated data, indicating high accuracy and precise parameter optimization. These algorithms effectively replicate the actual P-V characteristics, reflecting their robustness and reliability. TVETBO, while generally accurate, shows a slight deviation at higher voltages, suggesting it may require further refinement to achieve the same level of precision as the other algorithms. In Fig. 11D, while all algorithms show good alignment between actual and simulated data across most of the voltage range, there is a noticeable discrepancy at the peak power points. The simulated data tends to deviate from the actual data at the top of the curves, indicating that these algorithms may struggle to accurately capture the maximum power point behaviour of the

**Table 22 Summary of the best RMSE values found by the selected metaheuristics for the SDM, DDM, TDM, and PVMM.**

|         | SDM              | DDM              | TDM              | PVMM             |
|---------|------------------|------------------|------------------|------------------|
| COA(1)  | 9.8602187789E−04 | 9.8602187789E−04 | 9.8602187789E−04 | 1.2307306856E−02 |
| GOA     | 1.0914713470E−02 | 7.8737429493E−03 | 1.8837705005E−02 | 1.5384800935E−02 |
| COA(2)  | 2.8564228684E−03 | 3.5599768202E−03 | 3.6309167663E−03 | 1.5336896361E−02 |
| CPO     | 9.8614349211E−04 | 1.0025597044E−03 | 1.0046545920E−03 | 1.2307306856E−02 |
| GO      | 9.8602187789E−04 | 9.8248487610E−04 | 9.8248487610E−04 | 1.2307306856E−02 |
| APO     | 9.8602188389E−04 | 9.8478502518E−04 | 9.8463282698E−04 | 1.2307306856E−02 |
| MOA     | 6.1812674132E−03 | 1.1816280874E−02 | 5.2865429516E−03 | 1.2307306856E−02 |
| SBOA    | 1.1352185900E−03 | 1.0625737349E−03 | 1.0690313595E−03 | 1.2307306856E−02 |
| EOA     | 2.0449632490E−03 | 1.7907122489E−03 | 1.4307111957E−03 | 1.2340266929E−02 |
| TVETBO  | 4.1036065111E−02 | 3.9436054603E−02 | 4.5692730515E−02 | 2.7419994376E−02 |

**Table 23 Results of the Dunn's test using the RMSE values reported in Table 22.**

|        | GOA    | COA(2) | CPO    | GO     | APO    | MOA    | SBOA   | EOA    | TVETBO |
|--------|--------|--------|--------|--------|--------|--------|--------|--------|--------|
| COA(1) | 0.1631 | 0.7535 | 1.0000 | 1.0000 | 1.0000 | 0.8041 | 1.0000 | 0.9874 | 0.0221 |
| GOA    |        | 1.0000 | 0.5818 | 0.0342 | 0.1365 | 1.0000 | 0.8882 | 1.0000 | 1.0000 |
| COA(2) |        |        | 0.9932 | 0.3108 | 0.6986 | 1.0000 | 1.0000 | 1.0000 | 0.9999 |
| CPO    |        |        |        | 1.0000 | 1.0000 | 0.9966 | 1.0000 | 1.0000 | 0.1365 |
| GO     |        |        |        |        | 1.0000 | 0.3586 | 0.9997 | 0.7535 | 0.0033 |
| APO    |        |        |        |        |        | 0.7535 | 1.0000 | 0.9783 | 0.0177 |
| MOA    |        |        |        |        |        |        | 1.0000 | 1.0000 | 0.9997 |
| SBOA   |        |        |        |        |        |        |        | 1.0000 | 0.3586 |
| EOA    |        |        |        |        |        |        |        |        | 0.9647 |

PVMM. This discrepancy suggests that while the algorithms are generally reliable, their accuracy in predicting the PV module's performance at peak power requires improvement.

Table 22 provides a summary of the best RMSE values found by the selected metaheuristic algorithms for the different PV models, specifically the SDM, DDM, TDM, and PVMM. Table 23 summarizes the *p*-values obtained by the *post hoc* Dunn's test with a significance level set to 0.05. If the *p*-value is less than the chosen significance level, we would conclude that there are statistically significant differences among the algorithms' performances; otherwise, we would conclude that there are no statistically significant differences among the algorithms' performances. Table 24 illustrates the ranking of the optimizers considered for the comparative study using the Friedman test. COA(1) ranks second (2.75), suggesting it is one of the most effective algorithms, closely followed by CPO (3.875) and APO (2.625). GO, with the lowest rank of 1.75, emerges as the top-performing algorithm. On the other hand, TVETBO, ranked tenth (10), shows the least favourable performance among the algorithms considered. GOA (8.75), COA(2) (7.25), MOA

**Table 24  Results of the Friedman test using the RMSE values reported in Table 22.**

|       | COA(1) | GOA    | COA(2) | CPO    | GO     | APO    | MOA    | SBOA   | EOA    | TVETBO |
|-------|--------|--------|--------|--------|--------|--------|--------|--------|--------|--------|
| Ranks | 2.7500 | 8.7500 | 7.2500 | 3.8750 | 1.7500 | 2.6250 | 7.1250 | 4.6250 | 6.2500 | 10     |

(7.125), SBOA (4.625), and EOA (6.25) fall in the middle range, indicating moderate effectiveness.

## CONCLUSION AND PERSPECTIVES

In conclusion, this study has conducted an extensive comparative analysis of ten contemporary metaheuristic algorithms for parameter estimation of various PV models. This optimization problem requires precise parameter identification to capture the complex, nonlinear behaviours of PV cells, which are influenced by fluctuating environmental conditions. The estimation process is challenging due to high computational demands and potential optimization errors, both of which can impact the accuracy of performance predictions. Through detailed experimentation and performance evaluation across four distinct PV models—the SDM, DDM, TDM, and PVMM—we have demonstrated significant variances in the efficiency and accuracy of the algorithms. Notably, the GO has emerged as the most effective algorithm, as confirmed by the Friedman test rankings. This optimizer has achieved an RMSE of 9.8602187789E−04 for the single-diode model, 9.8248487610E−04 for both the double-diode and triple-diode models and 1.2307306856E−02 for the photovoltaic module model.

The results underscore the importance of selecting appropriate optimization algorithms tailored to specific PV models to achieve optimal performance. The comprehensive evaluation of execution time, number of function evaluations, and solution optimality provides a clear understanding of each algorithm's strengths and limitations. These insights are crucial for researchers and practitioners in the field of renewable energy, particularly in the development and optimization of solar PV systems.

Several avenues for future research and development can be considered:

- Combining the strengths of different metaheuristic algorithms to create hybrid optimization techniques could potentially yield even better performance and robustness in parameter estimation tasks.
- Extending the evaluation of these algorithms to real-world PV systems (*e.g.*, Photowatt-PWP201, STM6-40/36, STP6-120/36) and operational data would help in validating their practical applicability and effectiveness under varying environmental conditions.
- Refinement and adaptation of the existing algorithms, incorporating mechanisms to avoid local optima and improve convergence speed, can further enhance their performance.

### Funding

This work was supported by the Researchers Supporting Program (RSPD2024R809), King Saud University, Riyadh, Saudi Arabia. The funders had no role in study design, data collection and analysis, decision to publish, or preparation of the manuscript.

### Grant Disclosures

The following grant information was disclosed by the authors:
King Saud University, Riyadh, Saudi Arabia: RSPD2024R809.

### Competing Interests

The authors declare that they have no competing interests.

### Author Contributions

- Adel Zga conceived and designed the experiments, performed the experiments, performed the computation work, prepared figures and/or tables, authored or reviewed drafts of the article, and approved the final draft.
- Farouq Zitouni conceived and designed the experiments, performed the experiments, analyzed the data, performed the computation work, prepared figures and/or tables, authored or reviewed drafts of the article, and approved the final draft.
- Saad Harous conceived and designed the experiments, analyzed the data, prepared figures and/or tables, authored or reviewed drafts of the article, and approved the final draft.
- Karam Sallam conceived and designed the experiments, analyzed the data, performed the computation work, prepared figures and/or tables, authored or reviewed drafts of the article, and approved the final draft.
- Abdulaziz S. Almazyad conceived and designed the experiments, prepared figures and/or tables, authored or reviewed drafts of the article, and approved the final draft.
- Guojiang Xiong conceived and designed the experiments, prepared figures and/or tables, authored or reviewed drafts of the article, and approved the final draft.
- Ali Wagdy Mohamed conceived and designed the experiments, analyzed the data, prepared figures and/or tables, authored or reviewed drafts of the article, and approved the final draft.

### Data Availability

The third-party dataset to perform the comparative analysis of the 10 algorithms is available at: https://doi.org/10.1080/01425918608909835.

### Supplemental Information

Supplemental information for this article can be found online at http://dx.doi.org/10.7717/peerj-cs.2646#supplemental-information.

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
