# Peer review of "A comparative study of the performance of ten metaheuristic algorithms for parameter estimation of solar photovoltaic models"

_PeerJ Computer Science, doi:10.7717/peerj-cs.2646_

## Round 0.1 · original submission · Major Revisions

Dear Authors,
Your paper has been revised with interest but has not been recommended for publication in its current form. Please address the reviewers' concerns and resubmit your article once you have updated it. In particular:

1) You should insert more discussion in the abstract and conclusion sections, and the numerical data should support the findings.

2) You must motivate your study better, particularly if the problem domain is not clearly explained. Elaborate on the challenges encountered to accurately estimate the parameters of the PV cell/model using different modeling techniques. You must also explain why metaheuristic algorithms are considered promising solutions to address this optimization problem.

3) Considering that the novelty and contributions of the current study mentioned in Lines 80 to 86 are generic and vague, you must elaborate more on your research's significance.

Reviewer 1 ·

Basic reporting

See below

Experimental design

See below

Validity of the findings

See below

Additional comments

This paper presents a comparative study of the performance of ten metaheuristic algorithms for parameter estimation of solar photovoltaic models. The topic fits with the Journal of and the results are interesting. But the following comments need to be taken into account before the publication. I believe that the authors' response to the following major comments will positively affect the quality of the study.

* More discussion should be inserted in the abstract and conclusion sections and findings are supported with the numerical data.
*The introduction section is the most overlooked part of the study. Although the subject of the study is important, the authors could not reflect this importance to the reader. Please emphasize why this work is of vital importance in the introduction section. I I strongly encourage the authors to make further improvements to this chapter.
* It would also be useful to prepare a consolidated table summarizing existing literature studies in Section 3.
* Thoroughly check for grammar and refine the sentences for clarity and good understanding by others. There are grammar mistakes in the text, so it needs to improve the English language.
* The authors should compare their results with other existing literature studies such as https://doi.org/10.1016/j.envc.2023.100720,

* Table 3 should be revised to facilitate reader understanding.
* How do you ensure that meta-heuristic algorithms compete on fair terms when comparing search performance?

Reviewer 2 ·

Basic reporting

- Problem statement is missing in the abstract. Furthermore, more quantitative results should be reported in the abstract.
- Lines 57 to 64: It is necessary for authors to categorize the existing metaheuristic algorithms since there are many sources of inspiration used to develop these algorithms. Furthermore, authors should also introduce some recently proposed metaheuristic algorithms.
- The motivation of current study, particularly in the problem domain, are not clearly explained. Authors should elaborate the challenges encountered to accurately estimate the parameters of PV cell/model using different modeling techniques. It is necessary to provide such explanation and why metaheuristic algorithms are considered as promising solution to address this optimization problem.
- The novelty and contributions of current study that mentioned in Lines 80 to 86 are generic and vague. Authors are required to elaborate the significance of current study in more concrete manner.
- Pg 6 - It is more appropriate to present the values of I and V in the table instead of their current form.
- Paragraph 4 of Section 3 are much more lengthy than other paragraphs. Please organize the contents.
- Some relevant works such as those below are not reviewed:
(a) https://www.nature.com/articles/s41598-023-37824-4
(b) https://www.sciencedirect.com/science/article/pii/S2352484723011605

- Authors are required to summarize the papers presented in Section 3 in table format for better readability.

Experimental design

- It is important for the authors to justify why these particular 10 metaheuristic algorithms are chosen for the performance comparison study.
- The description of these 10 metahueristic algorithms are quite brief. Authors are required to elaborate their search mechanisms.
- It is crucial for the authors to provide a general workflow used to illustrate the application of these 10 metaheuristic algorithms. Currently, the problem formulation of PV estimation problem and the search mechanisms of algorithms are described as separate entities, and it is difficult to connect these two ideas together.

Validity of the findings

- Pg. 18 - What are the rationale of using the composite score to evaluate the performance of each configuration? Authors may need to provide further explanation about this. Please cite the relevant reference to support this.
- It is necessary for the authors to use boxplot to present the distributions of the results produced by each algorithm when they are used to solve the PV parameter estimation problem using different modeling technique.
- For convergence curves presented in Figures 7 to 10, authors should present them in the same graphs to better compare the convergence characteristics of each algorithm.

---

## Round 0.2 · Major Revisions

Dear Authors,
Your paper has been revised: although the majority of issues have been fixed, some concerns still need to be solved. In particular:

1) The literature review section on parameter estimation of solar cells using metaheuristic algorithms should be significantly improved
2) Using statistical analysis methods like Wilcoxon and Friedman would enrich the quality of your study.

Reviewer 1 ·

Basic reporting

In this round of revision, the author has made significant efforts to enhance the quality of the manuscript. The majority of the raised concerns have been effectively addressed. Nevertheless, I believe there are still opportunities for further enhancements in the manuscript. Therefore, I recommend another round of revision for this manuscript.

Experimental design

See below

Validity of the findings

See below

Additional comments

* The paper introduces a parameter estimation of various solar cells using metaheuristic algorithms. However, more details on the algorithm selection rationale would enhance clarity.
* The literature review part for parameter estimation of solar cells using metaheuristic algorithms should be significantly improved. Current literature studies should not be ignored.
* The use of statistical analysis methods like Wilcoxon and Friedman would enrich the quality of the study.

Reviewer 2 ·

Basic reporting

No comment

Experimental design

No comment

Validity of the findings

No comment

Additional comments

Authors have addressed all my comments raised in previous review. No further comments.

---

## Round 0.3 · accepted · Accept

Dear Authors,

Your paper has been accepted for publication in PEERJ Computer Science. Thank you for your fine contribution.

Reviewer 1 ·

Basic reporting

The quality of the manuscript improved significantly after the revision stage. The comments made by the authors are reasonable. I would like to thank the authors for making and explaining all necessary corrections in detail. The paper can be published in its current form.

Experimental design

The quality of the manuscript improved significantly after the revision stage. The comments made by the authors are reasonable. I would like to thank the authors for making and explaining all necessary corrections in detail. The paper can be published in its current form.

Validity of the findings

The quality of the manuscript improved significantly after the revision stage. The comments made by the authors are reasonable. I would like to thank the authors for making and explaining all necessary corrections in detail. The paper can be published in its current form.

Additional comments

The quality of the manuscript improved significantly after the revision stage. The comments made by the authors are reasonable. I would like to thank the authors for making and explaining all necessary corrections in detail. The paper can be published in its current form.